# Enhancing genome editing in hPSCs through dual inhibition of DNA damage response and repair pathways

Ju-Chan Park [1,2,6], Yun-Jeong Kim [1,2,6], Gue-Ho Hwang [3], Chan Young Kang[3], Sangsu Bae [3,4,5] & Hyuk-Jin Cha [1] ✉

Precise genome editing is crucial for establishing isogenic human disease models and ex vivo stem cell therapy from the patient-derived hPSCs. Unlike Cas9-mediated knock-in, cytosine base editor and prime editor achieve the desirable gene correction without inducing DNA double strand breaks. However, hPSCs possess highly active DNA repair pathways and are particularly susceptible to p53-dependent cell death. These unique characteristics impede the efficiency of gene editing in hPSCs. Here, we demonstrate that dual inhibition of p53-mediated cell death and distinct activation of the DNA damage repair system upon DNA damage by cytosine base editor or prime editor additively enhanced editing efficiency in hPSCs. The BE4stem system comprised of p53DD, a dominant negative p53, and three UNG inhibitor, engineered to specifically diminish base excision repair, improves cytosine base editor efficiency in hPSCs. Addition of dominant negative MLH1 to inhibit mismatch repair activity and p53DD in the conventional prime editor system also significantly enhances prime editor efficiency in hPSCs. Thus, combined inhibition of the distinct cellular cascades engaged in hPSCs upon gene editing could significantly enhance precise genome editing in these cells.

Since the advent of human induced pluripotent stem cells (iPSCs)[1], patient-derived iPSCs harboring pathogenic mutations have been established for modeling disease phenotypes[2,3]. Progress in genome editing technologies that permit precise gene correction of pathogenic mutations have enabled not only the establishment of isogenic pairs of patient-derived iPSCs but also ex vivo autologous stem cell therapy[4]. To this end, the efficient and precise correction of point mutations, which account for 58% of pathogenic genetic variants[5], has been of great interest. Knock-in (KI) of a desired sequence using Cas9-based homology-directed repair (HDR) has been extensively deployed in human pluripotent stem cells (hPSCs) [i.e., human embryonic stem cells (hESCs) and iPSCs][6,7]. However, the recent evidence that Cas9 endonuclease activity produces large unintended deletion mutations,

leading to chromosomal structure alteration[8–10] by the DNA double-strand break (DSBs) repair process[9], raises critical safety concerns for clinical applications. A recent study revealed that large, mono-allelic genomic deletions and loss-of-heterozygosity occur during HDR-mediated KI in up to 40% of iPSCs[11]. Since base editors (BEs) and prime editor (PE) technology developed from nickase Cas9 (nCas9) introduce the intended genetic variations without DSBs[12,13], the potential utility of BEs in clinical applications has received heightened interest[14]. Accordingly, BEs[15–20] and PE technology[21] have been applied in hPSCs for either disease modeling or mutation correction[22].

While DSBs introduced by Cas9 are mostly repaired by HDR or non-homologous end joining (NHEJ), site-specific cytosine deamination by cytosine base editors (CBE) or the synthesized 3′ DNA flap by

[1]College of Pharmacy, Seoul National University, Seoul, Republic of Korea. [2]College of Pharmacy and Research Institute of Pharmaceutical Sciences, Seoul National University, Seoul, Republic of Korea. [3]Genomic Medicine Institute, Seoul National University College of Medicine, Seoul, Republic of Korea. [4]Department of Biomedical Sciences, Seoul National University College of Medicine, Seoul, Republic of Korea. [5]Cancer Research Institute, Seoul National University College of Medicine, Seoul, Republic of Korea. [6]These authors contributed equally: Ju-Chan Park, Yun-Jeong Kim. ✉e-mail: hjcha93@snu.ac.kr

PE, is repaired by the base excision repair (BER)[23] or mismatch repair (MMR) pathway[24,25], respectively. Thus, transient inhibition of BER or MMR by expression of uracil DNA glycosylase inhibitor (UGI)[23,26] or dominant negative protein of MLH1 (dnMLH1), a key factor for forming the MMR complex[24], improves the editing efficiency of CBE or PE, respectively. In addition, not only Cas9[27] but also BEs[28] trigger the DNA damage response by activation of p53, often referred to as the guardian of the genome[29].

Unlike somatic cells or cancer cell lines, in which nascent genome editing tools are normally developed, hPSCs employ unique cellular defense mechanisms upon DNA damage to maintain their genome integrity[30]. They are highly sensitive to DNA damage triggered by p53-dependent mitochondrial cell death[31,32], and their DNA damage repair pathways are very active[33]. These distinct characteristics engender the different editing outcomes observed in these cells as reviewed[22]. DNA damage by Cas9 triggers prompt p53-mediated mitochondrial cell death in hPSCs[34]. When cultured in vitro, recurrent *TP53* loss of function (LoF) mutants favor survival[35], which results in the enrichment of p53 mutant hPSCs after Cas9-mediated genome editing[36]. Due to the highly active BER[37] and MMR in hPSCs[38] compared to differentiated somatic cells, depletion of uracil DNA glycosylase (UNG)[39] or MMR proteins[24] markedly enhances CBE or PE editing efficiency.

In the present study, we simultaneously inhibited two major cellular events, namely the p53-dependent DNA damage response and the specific DNA repair pathways (i.e., BER for CBE and MMR for PE) to improve the editing outcomes in hPSCs. Transient p53 inhibition with a dominant negative form of p53 (p53DD) favored hPSC cell survival induced by nickase Cas9 (nCas9) of BE4 or PE2. Thus, the dual inhibition vector system (i.e., AncBE4stem and PE4stem) readily achieves the additive effect on editing outcomes in hPSCs with no noticeable incidence of off-target effect, large deletion, and unintended transcriptome/genome-wide off-target effect.

## Results

### Dual inhibition of UNG and p53 improves CBE outcome

DNA modification with genome editing tools based on Cas9 inevitably leads to DNA damage, such as DSBs (by Cas9 endonuclease), single-strand breaks (SSBs) (by nCas9), base deamination (by CBE), and mismatch (by PE). These events trigger both the DNA damage response (e.g., cell cycle arrest, senescence, or apoptosis) and the DNA damage repair system (HDR, NHEJ, BER, and MMR) (Fig. S1A). Genome integrity is tightly maintained in hPSCs by their unique cellular characteristics (e.g., high susceptibility to DNA damage and active DNA damage repair)[30]. Because of these properties, genome editing in hPSCs engenders distinct outcomes compared to genome editing in somatic cells. Previously, we reported that poor CBE outcomes (i.e., C to T editing inefficiency and product impurity) result from high UNG activity in hPSCs[39]. Introduction of CBE simultaneously produces SSBs and G:U mismatch, respectively (Fig. 1A-a). The occurrence of SSBs activates p53 to initiate the DNA damage response (e.g., cell death especially in hPSCs) (Fig. 1A-b). Meanwhile, uracil in G:U mismatch is recognized and removed by UNG (Fig. 1A-c), forming an apurinic (AP) site (Fig. 1A-d) and subsequently resulting in G:C base repair (Fig. 1A-e) and production of C to G/A or indel mutations (Fig. 1A-f). Thereby, depletion of UNG with small interference RNA (siUNG) improves CBE editing outcome in hPSCs[39].

Thus, it was posited here that concurrent inhibition of the p53-dependent DNA damage response (leading to massive cell death in hPSCs) with p53DD and CBE-induced DNA damage repair with siUNG would improve CBE editing outcomes. The editing outcomes of AncBE4max with p53DD and siUNG (BE4 dual inhibition: BE4DI) (Fig. 1A, dotted box), were examined. The transient expression of p53DD significantly attenuated cell death upon expression of CBE (Fig. 1B, C). It is noteworthy that single cell dissociation-induced cell death of hPSCs, the distinct hPSCs' characteristic[40], requires the

treatment of Y-27632, a ROCK inhibitor[41] for electroporation, to prevent massive cell death (Fig. S1B). Due to the marginal effect of p53DD on the single cell dissociation-induced cell death of hPSCs (Fig. S1C), mock transfection, determined by EGFP expression, was not affected by p53DD (Fig. S1D), suggesting that p53DD expression rescues hPSCs upon DNA damage by AncBE4max.

In this context, C to T conversion efficiency of seven endogenous targets was significantly improved by BE4DI compared to individual inhibition (Fig. 1D). Unlike the sole inhibition of p53 with p53DD in CBE reported previously[42], BE4DI reduced the undesired editing outcomes such as C to G (Fig. 1E), C to A (Fig. S1E) and indel formation (Fig. S1F). Considering that innate UNG activity is responsible for the production of undesired editing outcomes (Fig. 1A-e, A-f), the enhancement of C to T conversion rate and product purity by BE4DI offers advantages over separate inhibition of p53 or UNG. Within the seven targets examined, HEK2 and HEK4 contain multiple cytosines within their respective editing windows, each exhibiting different levels of editing efficiency (Fig. S1G). Similarly, the dual inhibition strategy yielded an additive effect on bystander cytosines, specifically C4 in HEK2 and C8 in HEK4 (Fig. S1H, I).

### A single vector system for dual inhibition in CBE

For simple execution of dual inhibition of UNG and p53, a single vector system was designed from the AncBE4max system to harbor an additional UGI for inhibition of UNG (instead of siUNA) and p53DD, herein referred to as AncBE4stem (Fig. 2A). The editing outcomes of AncBE4stem were examined in six endogenous targets (Fig. 2B). As observed with BE4DI, introduction of a single vector of AncBE4stem significantly improved C to T conversion (Fig. 2B) with lesser incidence of unintended editing outcomes associated with CBE, including C to G (Fig. 2C), C to A (Fig. S2A) and indel formation (Fig. S2B). Statistical analysis underscored the improved editing outcome achieved by AncBE4stem (Fig. 2D). The effect observed with a single vector system on bystander cytosine, as demonstrated with BE4DI (Fig. S1G), was successfully replicated (Fig. S2C, D). The incidence of off-target effects at seven potential off-target sites (OT1 to OT4 at CCR5-3 and OT1 to OT3 at HEK2) was comparable to that of AncBE4stem (Fig. 2E). Notably, BE4max, known for inducing DSB and genotoxic byproducts such as large deletion in hematopoietic stem cells (HSCs), largely associated to inhibition of UNG in BE4max[43] promoted a closer examination of AncBE4stem's impact on large deletion in hPSCs. Utilization of nanopore long-read sequencing techniques[44,45], the assessment of large deletion (deletion more than 100 bp long) revealed a low but comparable level with dual inhibition (Fig. 2F and S3E). To further explore the effect of the dual inhibition on guide RNA (gRNA) independent off-target effects, genome-wide C to T and transcriptome-wide C to U conversion (Fig. 2G) was monitored along with the outcome of base editing on HEK3 (Fig. S2F). According to the counts of nucleotide conversions, both genome-wide (C to T) and transcriptome-wide (C to U) conversions, showed comparable patterns between AncBE4-max and AncBE4stem (Fig. 2G).

### Dual inhibition of MMR and p53 improves PE outcome

As is true of CBE, SSBs induced by nCas9 activity in PE, which precipitates the DNA damage response, occur near PAM sequences (Fig. 3A-a) along with the synthesized DNA strand containing the desired edit (3' flap) via reverse transcriptase (RT) (Fig. 3A-b). The intermediate product formed by annealing of the 3' flap and 5' flap excision (Fig. 3A-c) is inserted at the target site by ligation and consequent DNA replication (Fig. 3A-d). Innate MMR activity recognizes the intermediate product (Fig. 3A-c) and removes it (Fig. 3A-e). Transient inhibition of MMR activity by dnMLH1, a key component for MutL complexes, enhances PE outcomes[24]. A recent study investigated the likelihood of PE efficiency improvement by p53DD expression in hPSCs[42], which should result from increased survival due to inhibition

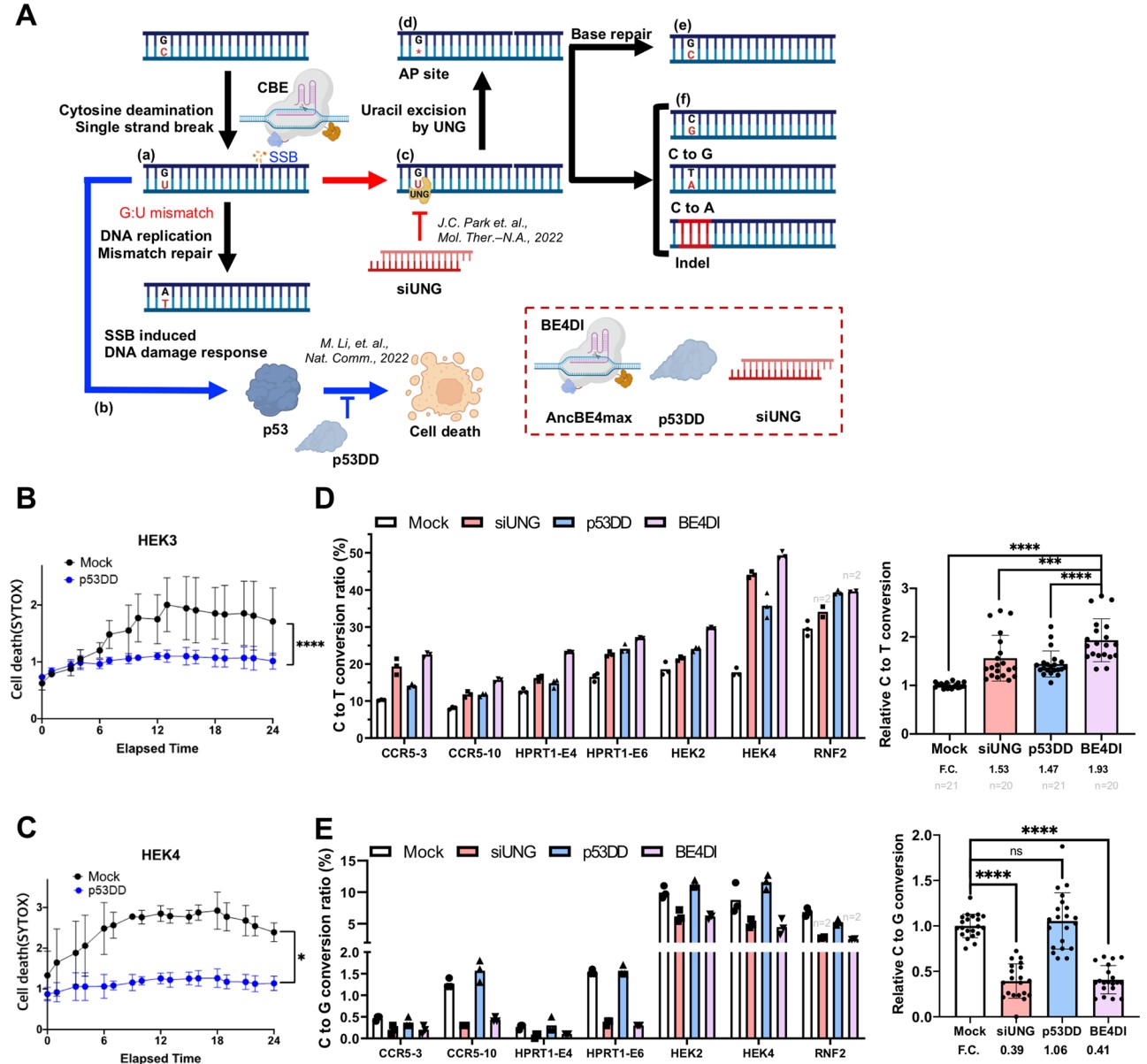

**Fig. 1 | Dual inhibition of UNG and p53 improve CBE outcome. A** Scheme of DNA damage response and DNA repair pathways followed by cytosine base editor (CBE), DNA single-strand break (SSB) induced pathway is colored in blue, and uracil DNA glycosylase (UNG) mediated uracil excision process is colored in red. Created with BioRender.com. **B**, **C** Cell death after gene editing was tested by live-cell imaging and SYTOX staining. The imaging was started after 24 h of transfection of gRNA for HEK3 (**B**) or HEK4 (**C**) and AncBE4max without Y27632 treatment. **D**, **E** C to T substitution (**D**) and C to G substitution (**E**) ratio in H9-hESCs after AncBE4max plasmid delivery with non-targeting siRNA and pcDNA 3.0 vector (Mock), siRNA targeting UNG (siUNG), p53DD expression vector (p53DD), and both siUNG and p53DD (BE4DI), for the left panel, $n = 3$ except the designated replicates. $n$ always represents the biologically independent samples if not else described. Bars represent mean values, and error bars represent the S.D. of independent biological replicates. Detailed information on statistical analysis is listed in the "Statistical analysis" section. The source data of **B**–**E** are provided in the Source Data file.

of the p53-dependent DNA damage response (e.g., cell death) (Fig. 3A). As predicted, p53DD expression in parallel with introduction of PE2 increased surviving colony (Fig. 3B) through inhibition of cell death upon PE2 expression (Figs. 3C and S3A), which would account for increase of editing efficiency (Fig. S3B). Accordingly, p53DD expression with PE4 (PE2 with dnMLH1) or PE5 (PE3 with dnMLH1), achieving dual inhibition for prime editing, clearly enhanced the editing efficiency (Fig. 3D). A single vector system, introducing p53DD on PE4 or PE5 (i.e., PE4stem or PE5stem respectively) for dual inhibition of the DNA damage response with p53DD and PE-specific DNA damage repair (i.e., MMR) with dnMLH1 (Fig. 3E), noticeably improved the PE outcomes from single nucleotide conversion (T to A), deletion (1–5 del

and 1–15 del), three nucleotide insertion (GTA ins) and 38 nucleotide insertion (AttB ins) (Fig. 3F). Statistical analysis of relative editing efficiency highlighted the effect of dual inhibition in PE4 (Fig. 3G) and PE5 (Fig. 3H), with no noticeable off-target editing (Fig. 3I). PE induced lowered but not abrogated induction of large deletion than Cas9 in HSCs[44]. As similar as the result in CBE, neither notable change of large deletion (more than 100 bp long) (Fig. 3J and S3C) was observed by dual inhibition. Additionally, the comparison of gRNA-independent off-target effects on the transcriptome and genome revealed that the outcomes were comparable between PE4max and PE4stem (Fig. 3K), along with PE editing on HEK3 (Fig. S3D). To further closely examine potential gRNA-dependent off-target effects in hESCs, single edited

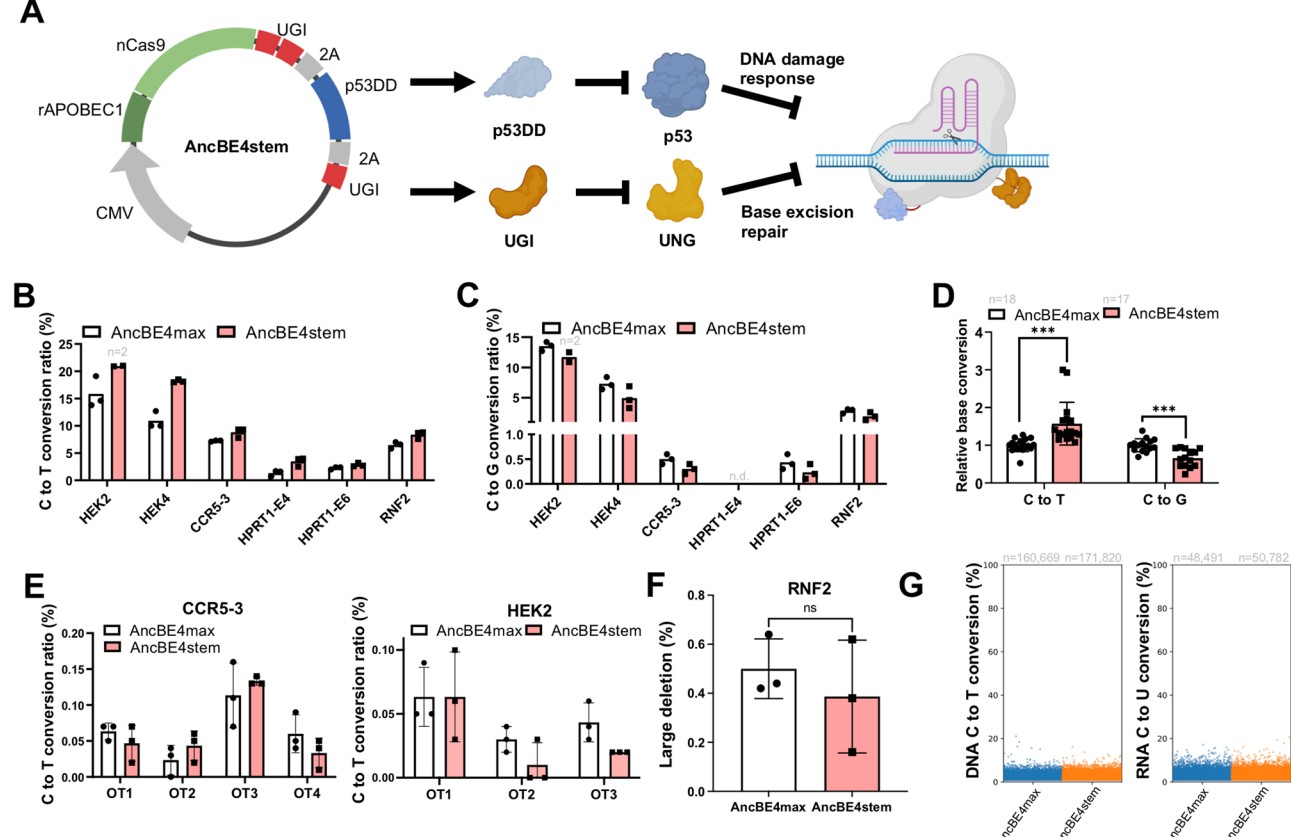

**Fig. 2 | A single vector system for dual inhibition in CBE. A** Scheme of AncBE4stem construction, created with BioRender.com. **B–D** C to T (**B**), C to G (**C**) conversion efficiency, and total relative base conversion (**D**) ($n = 3$ except the designated replicates, n.d. for not detected data) of AncBE4max and AncBE4stem at the indicated target sites. **E** Off-target editing of CCR5-3 targeted sites (OT1, OT2, OT3 and OT4) and HEK2 targeted sites (OT1, OT2 and OT3) with AncBE4max and AncBEstem at the indicated target sites ($n = 3$). **F** Deletion more than 100 base pair long at RNF2 target treated with AncBE4max and AncBE4stem analyzed by

Nanopore long-read sequencing ($n = 3$). **G** A jitter plot of the DNA C to T substitution and RNA C to U from whole-exome sequencing data and transcriptome data of AncBE4max and AncBE4stem treated cells with HEK3 targeting gRNA ($n = 1$). $n$ always represents the biologically independent samples if not else described. Bars represent mean values, and error bars represent the S.D. of independent biological replicates. Detailed information on statistical analysis is listed in the "Statistical analysis" section. The source data of B-F are provided in the Source Data file.

clones (Edit#1 and Edit#2) were obtained and compared with an unedited single clone (Mock) (Fig. S3E). Notably, neither gRNA-dependent off-target effects (Fig. S3F, G) nor abnormalities in karyotyping (Fig. 3L) were detected in the edited single clones (RNF2 GTA ins and HEK3 1–10 del) relative to the Mock. Moreover, the edited single clones preserved normal pluripotency, as evidenced by the expression of standard pluripotent marker genes (Fig. 3M) and alkaline phosphatase activity (Fig. S3H).

### Dual inhibition approach in p53 mutant hPSCs

Correction of pathogenic mutations in patient-derived hPSCs using BEs or PE transforms edited hPSCs into promising cell sources for autologous ex vivo stem cell therapy[22]. Nevertheless, the recurrent genetic alterations such as p53 LOF mutations, which confer a survival advantage, lead to the dominance of mutant hPSCs during in vitro culture[35]. Thus, p53 mutant hPSCs, when co-exists with normal hPSCs, are enriched in the surviving clones after Cas9 treatment due to the acquired resistance to cell death[36].

The investigation into the competition-winning ability of p53 mutant hPSCs in gene correction was conducted with iPSCs (SES8, established by lentiviral reprogramming[46]) harboring a p53 mutation at R175, which corresponds to the recurrent p53 mutant site in hPSCs[35] (Fig. 4A). The impact of p53DD on cell death induced by PE2 was found to be only marginal (Fig. 4B). Similar result was obtained with another

p53 mutant iPSC line (BJ-iPSCs, established by episomal reprogramming[47]) with normal karyotyping (Fig. S4A). These p53 mutant iPSC lines were free from mycoplasma contamination (Fig. S4B, C). The diminished survival benefit conferred by p53DD in p53 mutant iPSCs (Fig. 4B and S4D), halved the gene-editing outcomes by dual inhibition approach. The effect of UNG inhibition (with siRNA for UNG) in BE4DI was solely achieved in C to T conversion (Fig. 4C, D), and the unintended CBE outcomes such as C to G (Fig. 4E and S4E), C to A (Fig. S4F) and indel formation (Fig. S4G). Similar results were obtained with AncBE4stem for C to T (Fig. 4F), C to G (Fig. 4G and S4H), C to A (Fig. S4I), and indel formation (Fig. S4J). Of interest, the impact of MMR inhibition with dnMLH1 persisted in p53 mutant hPSCs (Fig. 4H), despite only marginal effect of p53DD in PE4 and PE5 (Fig. 4I, J).

Subsequently, the enrichment of p53 mutant hPSCs was assessed following the introduction of PE2 in the co-culture with normal and p53 mutant lines (Fig. S5A). Consistent with the previous report[36], the surviving clones after PE2 revealed a significant enrichment of p53 mutant hPSCs (Fig. 4K). We next hypothesized that providing a transient survival benefit to normal hPSCs with p53DD would enable them to compete with p53 mutants in the presence of DNA damage induced by PE2, eventually inhibiting the enrichment of p53 mutant hPSCs during the gene-editing process. Unfortunately, despite multiple attempts the desired effect of p53DD in inhibiting the enrichment of

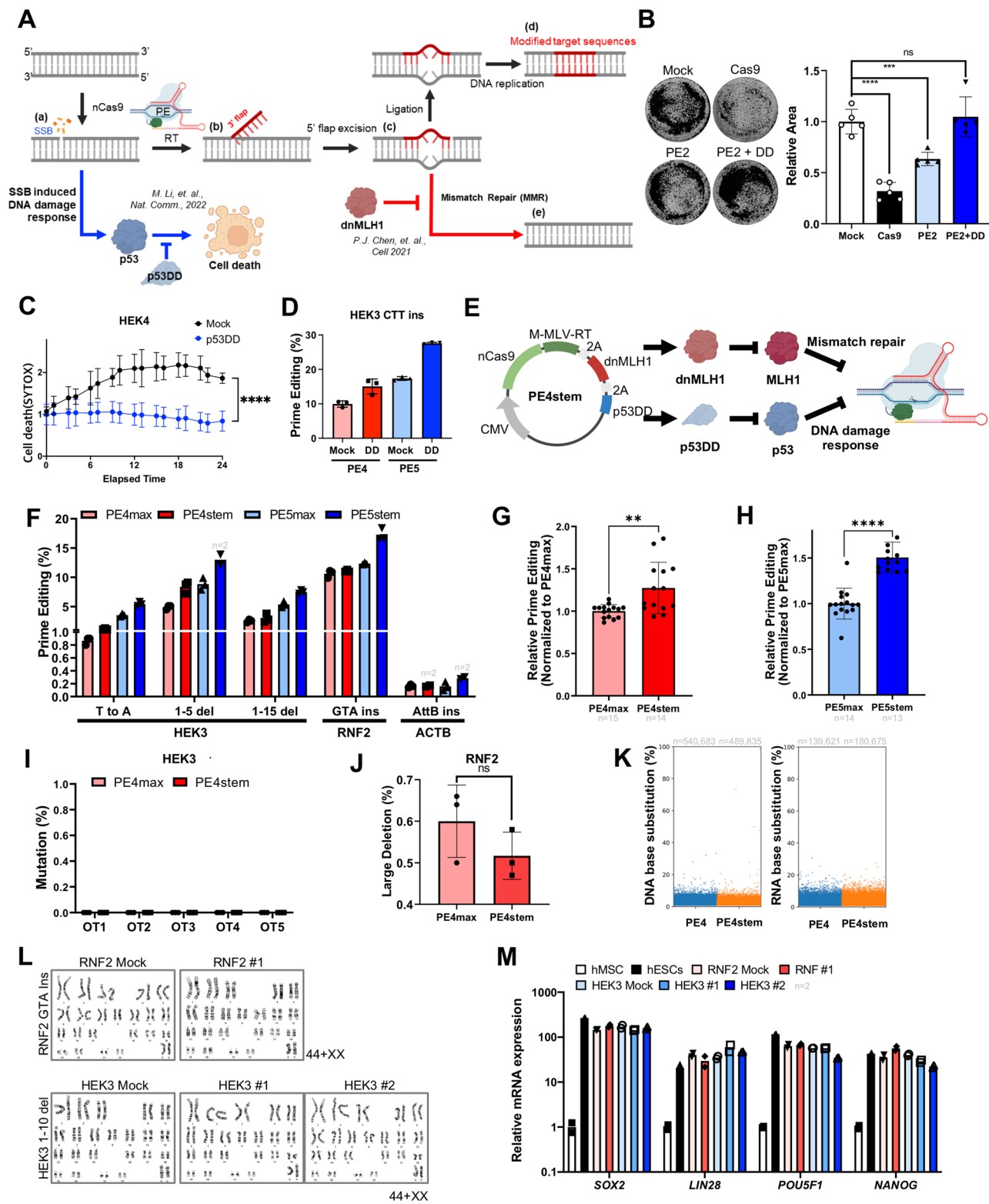

p53 mutant hPSCs after PE2-mediated prime editing was not achieved (Fig. S5B).

## Discussion

Recent research has drawn attention to the impact of the DNA damage response and repair pathways on the efficiency of precise genome editing tools like CBE and PE in hPSCs[39,48]. Informed by these findings, we've demonstrated in the present study that concurrent inhibition of both the DNA damage response and DNA repair pathways represents a straightforward strategy to enhance editing outcomes. Notably, similar results were obtained using one-vector systems incorporating p53DD and respective components to disrupt the DNA repair system, such as AncBE4stem derived from AncBE4max[49] and PE4stem derived from PE4max[24]. In our hands, these approaches proved particularly effective in hPSCs, where editing outcomes are typically poorer than in somatic cancer cell lines. This is partly due to high activity of p53-mediated cell death and active DNA damage repair mechanisms in hPSCs. Precise genome editing tools like BEs and PE enable the

**Fig. 3 | Dual inhibition of MMR and p53 improve PE outcome. A** Scheme of the prime editing process, the DNA single-strand break (SSB) response is colored in blue, mismatch repair (MMR) process is colored in red. Created with BioRender.com. **B** Image of crystal violet assay and relative areas (normalized by Mock) in H9-hESCs after transfection of Cas9 without gRNA (Mock), Cas9, PE2, and PE2 with p53DD vector (PE2 + DD). **C** Cell death after gene editing was tested by live-cell imaging and SYTOX staining. The imaging was started after 24 h of transfection of pegRNA for HEK4 ATCG insertion and PE4. **D** Prime editing efficiency for HEK3 CTT insertion by PE4 and PE5(Mock), PE4 and PE5 with p53DD expression (DD) in H9-hESCs ($n = 3$). **E** Scheme of PE4stem construction, created with BioRender.com. **F** Prime editing efficiency for indicated target by PE4max, PE4stem, PE5max, and PE5stem ($n = 3$ except the designated replicates). **G, H** Relative prime editing efficiency of PE4max and PE4stem (**G**) and PE5max and PE4stem (**H**), **I** Off-target editing of HEK3 1–5 deletion at off-target sites (OT1, OT2, OT3, OT4 and OT5) by PE4max and PE4stem. ($n = 3$). **J** Deletion more than 100 base pair long at RNF2 target treated with PE4max and PE4stem analyzed by Nanopore long-read sequencing ($n = 3$). **K** A jitter plot of the DNA and RNA base substitution from whole-exome sequencing and transcriptome data of PE4max and PE4stem treated cells with HEK3 CTT insertion pegRNA ($n = 1$). **L** Karyotype of unedited (RNF2 Mock and HEK3 Mock) and edited single clones (HEK3 #1, HEK3#2 and RNF1#1) of H9-hESCs (44 + XX). **M** The mRNA levels of pluripotency marker genes in hESCs, unedited (RNF2 Mock and HEK3 Mock) and edited clones (HEK3 #1, HEK3#2 and RNF1#1), human mesenchymal stem cells (hMSCs) as a negative control ($n = 2$). $n$ always represents the biologically independent samples if not else described. Bars represent mean values, and error bars represent the S.D. of independent biological replicates. Detailed information on statistical analysis is listed in the "Statistical analysis" section. The source data of **B–D**, **F–J**, and **M** are provided in the Source Data file.

correction of pathogenic mutations, and these tools are continually advancing, opening up possibilities for ex vivo stem cell therapy derived from patient-specific iPSCs[22]. Ensuring the safety of precise gene editing is of utmost importance in a clinical context. Therefore, it is crucial to extensively evaluate the effect of the editing toolkits, which have primarily been developed in somatic cancer cell lines, on hPSCs. In in vitro culture, p53 mutants have been observed to escape p53-dependent cell death[35], surviving p53-dependent cell death upon Cas9-induced DNA damage[34]. This accounts for the enrichment of p53 mutants upon clonal selection by Cas9[36] as well as by PE (Fig. 4K). The strategy for conferring the transient survival benefits to normal counterparts using p53DD failed to yield the intended effect (Fig. S5B). Given the challenges in inhibiting p53 mutant enrichment, assessing the p53 status of corrected hPSCs prior to further clinical usage for autologous stem cell therapy is advisable.

It is important to consider that the DNA damage response, encompassing p53-mediated cell death and DNA damage repair, may vary by cell type. For instance, cell lines like 293 T (largely used for the development of editing toolkits[50]) HCT116, DLD1, HCT15, and SW48 are deficient in mismatch repair (MMR)[51,52], rendering the inhibition of MMR minimally impactful on PE editing outcomes in these particular cell types. On the other hand, p53 status, frequently associated with mutations in many cancer cell lines (https://tp53.isb-cgc.org/), should be taken into consideration before employing dual inhibition strategies. For example, 293T, K562, and HCC1954, frequently used for the initial test of editing toolkits[12,50,53] are either p53-inactive (due to T antigen in 293T) or p53 mutant.

Consequently, the inhibition of p53 emerges as the most effective approach for PE and CBE in cell types highly susceptible to p53-mediated apoptosis following DNA damage. In particular, HSCs, the exclusive cell source for autologous ex vivo stem cell therapy and the first-in-kind FDA-approved gene-editing therapy product (i.e., CASGEVY), exhibit heightened vulnerability to cell death following DNA damage, primarily due to p53 activation[54,55] and compromised DNA damage repair[56], which leads to the accrual of DNA damage over time[57]. These distinct characteristics of HSCs may underlie the unexpected genotoxic outcomes of CBE and PE in HSCs[43]. In this context, the sole inhibition of p53 using p53DD represents the optimal strategy for therapeutic gene editing of HSCs in conditions such as sickle cell anemia[58] or β-thalassemia[59], both of which are subjects of ongoing clinical trials (NCT05456880). The encouraging outcomes in gene-corrected HSCs with base editing have extended the applicability to other cell types such as hepatic progenitors[60], keratinocytes[61], and so on. Therefore, similar to the case of hPSCs and HSCs, a comprehensive understanding of the DNA damage response and repair capacity of the target cell types needs to be considered before implementing BEs or PE.

In conclusion, we have developed a streamlined one-vector system for simultaneous inhibition of the DNA damage response following CBE or PE, leading to improved editing outcomes in hPSCs.

## Methods

### Ethical statement
All hESCs and iPSCs experiments were performed at the Seoul National University and followed the 2016 Guidelines for Stem Cell Research and Clinical Translation released by the International Society for Stem Cell Research (ISSCR). hESCs and iPSCs work was reviewed and approved by the Institutional Review Board at Seoul National University (SNU IRB protocol #2305/003-014).

### Plasmid construction
p3s-Cas9-HN (addgene # 104171, from Dr. Jin-Soo Kim), pCMV CBEmax (addgene # 119801, from Dr. David Liu), pCMV-AncBE4max (addgene # 112094, from Dr. David Liu), T7-p53DD-pcDNA3 (addgene # 25989, from Dr. William Kaelin), AncBEStem and PE4stem plasmids were used in this research. BE4stem and PE4stem were constructed by Gibson cloning using Gibson Assembly Master Mix (NEB #E2611). Additional UGI in BE4stem was amplified from pCMV-BE3 (addgene # 73021, from Dr. David Liu) and p53DD in AncBE4stem and PE4stem was amplified from T7-p53DD-pcDNA3 (addgene # 25989) by PCR. PCR was performed with a KOD Multi & Epi PCR kit (TOYOBO).

### Cell culture and transfection
H9 (WA09, WiCell Research Institute) hESC and iPSCs (BJ-iPSCs) were grown in StemMACS media (Miltenyi-Biotec) containing 50 mg/mL Gentamicin on dishes coated with Matrigel (BD Biosciences). 200 mL of Matrigel was diluted in 16 mL of chilled DMEM/F-12 fluid for the Matrigel coating (Gibco). A culture plate was doused with diluted Matrigel and then incubated for one hour in a cell culture incubator. Dulbecco's phosphate-buffered saline (DPBS) was used to cleanse and separate hESCs before the transfer (#561527, BD Biosciences). Three rounds of DMEM/F-12 media washing were performed on detached cells. After washing, the cells were resuspended in 1 mL of StemMACS medium and then plated on a Matrigel-coated plate (Gibco). According to the standardized protocol for electroporation of hPSCs, gene delivery with the electroporator (NEPA-21, NEPAGENE), hPSCs were rinsed with DPBS and detached with Accutase (#561527, BD Biosciences). Cells were resuspended and diluted to $1 \times 10^6$ cells per 100 μL with Opti-MEM (#31985070, Gibco) after three rounds of Opti-MEM washing (#31985070, Gibco). 3 μg of the sgRNA or pegRNA vector and 2 μg of the Cas9, BE, or PE vectors were added to the cell solution. Additional 2 μg of siUNG and/or p53DD overexpression vector were added to the cell mixture for siUNG and/or p53DD overexpression vector, respectively. The same amount of siNC and pcDNA 3.0 was added to the cell solution for the control of siUNG and p53DD respectively. The electroporation was performed with NEPA-21 electroporator with 175 V, 2.5 ms of poring pulse as described[20]. Dissociated cells after electroporation were plated into a culture dish with 10 μM of Y-27632.

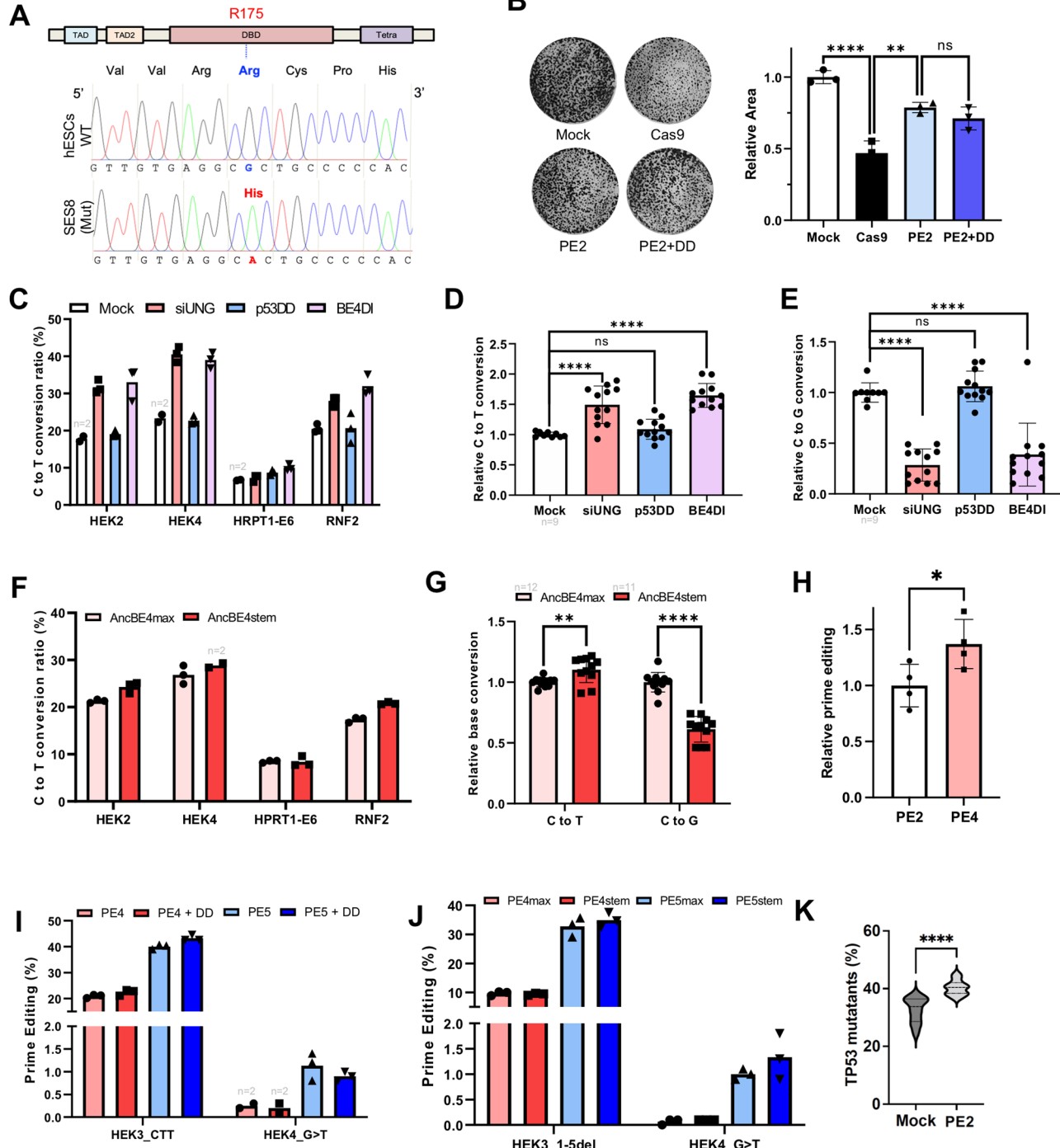

**Fig. 4 | Dual inhibition approach in p53 mutant hPSCs. A** Sequences in 175R (Arginine) region of WT hESCs and TP53 mutant iPSCs (SES8). The amino acid of mutated SES8 was altered to H (Histidine) by one-base substitution. **B** Colony area calculated after Cas9 and PE gene editing in TP53 mutant iPSCs (SES8). Colony area calculated by ImageJ after gene editing in SES8. **C** C to T substitution ratio of each indicated target by AncBE4max plasmid with non-targeting siRNA and pcDNA 3.0 vector (Mock), siRNA targeting UNG (siUNG), p53DD expression vector (p53DD), and both siUNG and p53DD (BE4DI) at 4 genomic sites in SES8 (*n* = 3 except the designated replicates). **D, E** Normalized C to T (**D**) and C to G substitution (**E**) of Fig. 4C (*n* = 12 except the designated replicates). The C to T and C to G substitution ratio is normalized to Mock efficiency. **F, G** C to T conversion ratio (**F**) and total relative C to T and C to G (**G**) of AncBE4max and AncBEstem in SES8 (*n* = 3 except the

designated replicates). **H** Relative prime editing efficiency of PE2 and PE4 (*n* = 4). **I, J** Prime editing efficiency of PE4max (PE4), PE4max with p53DD (PE4 + DD), PE5max (PE5) and PE5max with p53DD (PE5 + DD) (**I**) and PE4max, PE4stem, PE5max, and PE4stem (**J**) of indicated target (*n* = 3 except the designated replicates). **K** PE2 effects on the enrichment of *TP53* mutant population. WT hPSCs and TP53 mutant hPSCs were mixed in a certain ratio, and then PE2 and pegRNA vectors were treated for gene editing. After 3 days, gDNA for each sample was harvested and the ratio of R175 mutation was tested by NGS. n always represents the biologically independent samples if not else described. Bars represent mean values, and error bars represent the S.D. of independent biological replicates. Detailed information on statistical analysis is listed in the "Statistical analysis" section. The source data of **B**–**K** are provided in the Source Data file.

## Targeted deep sequencing

At three days following transfection, genomic DNA was extracted from Cas9-, BE-, or PE-transfected cells using a Wizard Genomic DNA Purification Kit (#A1120, Promega) for the purpose of analyzing editing effectiveness. A KOD Multi & Epi PCR kit (TOYOBO) was used to amplify the target sites in order to create the sequencing library. These libraries were sequenced utilizing MiniSeq using an Illumina TruSeq HT Dual Index system. In a nutshell, the Illumina MiniSeq platform was used to perform paired-end read sequencing on an equal number of PCR amplicons. Following MiniSeq, paired-end reads were examined by utilizing BE-analyzer[62] and PE-analyzer[63] to analyze genome editing outcomes. The targeted deep sequencing was provided by Macrogen. Inc.(https://www.macrogen.com/ko/main).

## Editing efficiency averaging and normalization

The arithmetic mean of each experiment was used to compute the average value of editing efficiency for each target. The arithmetic mean of the individual targets was used to compute the average value of editing efficiency in each cell line. By dividing the average value of the editing efficiency of each target from perturbation (such as siUNG or p53DD) by that of the control, editing efficiency was normalized.

## Live-cell imaging and cell death assay

Concerning all of the bright field images captured, a light channel optical microscope (CKX-41, Olympus, Tokyo, Japan) or JULI-stage (NanoEntek, Seoul, Korea) was used in accordance with the manufacturer's protocol. Cell death after electroporation was analyzed by JuLi-Stage live-cell imaging with SYTOX (#S7020, Thermofisher) staining and analyzed by JuLi-Stat as per the manufacturer's manual.

## Fluorescence-based competitive proliferation assay

EGFP-expressing hESC and TP53 mutant hESC were cultured together. Cells were detached with Accutase (#561527, BD Bioscience) and rinsed with DPBS three times before flow cytometry. EGFP+ cells in the total population were measured using flow cytometry.

## Whole-exome sequencing

The gDNA sample for Whole-exome sequencing was harvested by using Wizard® Genomic DNA Purification Kit (Promega). Target enrichment and sequencing to generate a gDNA library were prepared from 50 ng input of gDNA using the Twist Library Preparation EF kit (96 samples, PN 101058) and TruSeq-compatible Y-adapters (Ilumina). The DNA quality and quantity were measured by PicoGreen and agarose gel electrophoresis. After the gDNA fragmentation, end-repair and addition of "A" tail were performed, followed by PCR amplification. The final product was quantified by TapeStation DNA screentape D1000 (Agilent) and PicoGreen. The Ilumina platform generates raw images and base calling with an integrated primary analysis software RTA. FASTQ files were generated by Illumina package bcl2fastq v2.20.0. The sequencing quality was validated by FastQC.

## RNA-seq and whole-exome sequencing data analysis

The data results of RNA-seq and whole-exome sequencing were aligned to the Human Reference Genome (GRCh38) using BWA-mem with the default option. The alignment results were sorted using samtools. For detailed analysis of substitutions, REDItools2 (https://github.com/BioinfoUNIBA/REDItools2) was applied as described previously[64]. Substitutions were identified based on the following criteria: (i) The depth of coverage at the substitution site exceeds 10 in both CRISPR-treated and control samples, (ii) The substitution frequency surpasses 1% in the CRISPR-treated sample, and (iii) The substitution frequency remains below 1% in the control sample. These substitutions were analyzed in two distinct contexts: (i) C to T substitutions in samples treated with BEs; and (ii) all types of substitutions

in samples treated with PEs. The plots were generated in Python using matplotlib and seaborn module.

## Large deletion analysis using Nanopore

The targeted regions were amplified using long-range PCR with KOD multi & epi DNA polymerase (TOYOBO, KME-101). The primers used in long-range PCR were designed using Primer-BLAST. The PCR products were purified using AMPure XP beads (BECMAN COULTER, #A63881). The purified samples, quantified to 200 fmol, were used for Nanopore library preparation. The Nanopore library was prepared for R10.4.1 Minion Flow cell (Oxford Nanopore Technologies, FLO-MIN106D) using Ligation Sequencing Kit V14 (Oxford Nanopore Technologies, SQK-LSK114). The prepared library was sequenced on the MinION sequencer (Oxford Nanopore Technologies, Mk1B) using the 260 bps option. To generate FATSQ files, the sequencing outputs were processed using guppy basecaller, employing the super high accuracy module (dna_r10.4.1_-e8.2_260bps_sup.cfg). The FASTQ files were aligned to the reference sequence using a guppy aligner with the default option. The alignment results were analyzed using Python. The mutations were classified into WT, deletions, insertions, and large deletions based on the length of mutations that spanned the cleavage sites±100 bp. The Python codes were uploaded to GitHub (https://github.com/Gue-ho/STEM_BE_PE_LD_Analysis).

## Statistical analysis

The mean values and standard deviation are used to express the quantitative data (SD). Statistical analysis was performed using the GraphPad Prism 10.2.2. Software (SanDiego, CA). For hypothesis testing, analysis of variance(ANOVA), and Student's $t$-test were used respectively. In order to be considered statistically significant, values had to be less than 0.05 (*$p < 0.05$, **$p < 0.01$, ***$p < 0.001$, and ****$p < 0.0001$, 95% CI). The non-significant comparison was annotated as "ns". One-way ANOVA was used for the comparison of the conversion ratio between multiple groups (Fig. 1D, E and S1E, F, S3B, 4B, 4D, 4E, S5B). Two-way ANOVA was used for the comparison of cell death between Mock and p53DD expression group (Fig. 1B, C and S1B, C, 3C, S3A) in a time-dependent manner. Student's $t$-test (two-tailed) was used for the comparison of two unpaired groups (S1D, G, 2D, 2F, 3G, 3H, 3J, 4G, 4H, 4K, S4D). Post-hoc test $p$-values in figures have been corrected for multiple comparisons via the Prism recommendation (Tukey or Dunnett) method.

For each figure, the numbers ($n$) of samples in each experimental group were as follows:

Figure 1B: $n = 3$ biologically independent samples for Mock and p53DD. Two-way ANOVA; Time x Column Factor $F_{(16,32)} = 4.877$, $p < 0.0001$. C: $n = 3$ biologically independent samples for Mock and p53DD. 2-Way ANOVA; Time x Column Factor $F_{(16,64)} = 1.843$, $p = 0.0441$/Right panel of D: $n = 21$ biologically independent samples for Mock and p53DD, $n = 20$ biologically independent samples for siUNG and BE4DI. One-way ANOVA, $p < 0.0001$; Dunnett's post-hoc test $p$-values corrected for multiple comparisons: Mock vs. siUNG Adjusted $p < 0.0001$, Mock vs. p53DD Adjusted $p = 0.0003$, Mock vs. BE4DI Adjusted $p < 0.0001$./Right panel of E: $n = 21$ biologically independent samples for Mock and p53DD, $n = 20$ for siUNG and BE4DI. One-Way ANOVA Adjusted $p < 0.0001$; Dunnett's post-hoc test $p$-values corrected for multiple comparisons: Mock vs. siUNG Adjusted $p < 0.0001$, Mock vs. p53DD Adjusted $p = 0.8270$, Mock vs. BE4DI Adjusted $p < 0.0001$.

Figure S1B: $n = 3$ biologically independent samples for Mock and p53DD, 2-Way ANOVA; Time x Column Factor $F_{(14,56)} = 21.17$, $p < 0.0001$./Right panel of D: $n = 12$ biologically independent samples for Mock and p53DD. Unpaired Student's $t$-test (two-tailed), $p = 0.5309$/Right panel of E: $n = 20$ biologically independent samples

for p53DD and Mock, $n = 19$ biologically independent samples for siUNG and BE4DI. One-way ANOVA, $p < 0.0001$; Dunnett's post-hoc test $p$-values corrected for multiple comparisons: Mock vs. siUNG Adjusted $p = 0.1123$, Mock vs. p53DD Adjusted $p = 0.0255$, Mock vs. BE4DI $p = 0.0151$ /Right panel of F: $n = 21$ biologically independent samples for Mock and p53DD, $n = 20$ biologically independent samples for siUNG and BE4DI. One-way ANOVA, Adjusted $p < 0.0001$; Dunnett's post-hoc test $p$-values corrected for multiple comparisons: Mock vs. siUNG $p < 0.0001$, Mock vs. p53DD Adjusted $p = 0.7977$, Mock vs. BE4DI Adjusted $p < 0.0001$ /Right panel of G: $n = 3$ biologically independent samples for Mock and p53DD of HEK2, unpaired Student's $t$-test (two-tailed), $p = 0.0065$ & $n = 3$ biologically independent samples for Mock and p53DD of HEK4, unpaired Student's $t$-test, $p < 0.0001$

Figure 2D: $n = 18$ biologically independent samples for AncBE4-max, $n = 17$ biologically independent samples for AncBE4stem. Unpaired Student's $t$-test (two-tailed), $p = 0.0002$ for C to T and $0.0001$ for C to G./F: $n = 3$ biologically independent samples for AncBE4max and AncBE4stem. Unpaired Student's $t$-test (two-tailed), $p = 0.2378$.

Figure 3B: $n = 5$ for every group, one-way ANOVA $p < 0.0001$; Tukey's post-hoc test $p$-values corrected for multiple comparisons: Mock(Cas9 only) vs. Cas9 Adjusted $p < 0.0001$, Mock (Cas9 only) vs. PE2 Adjusted $p = 0.0017$, Mock (Cas9 only) vs. PE2 + DD Adjusted $p = 0.9373$. **C:** $n = 3$ for Mock and p53DD. Two-way ANOVA; Time x Column Factor $F_{(16,64)} = 28.65$, $p < 0.0001$./G: $n = 12$ biologically independent samples, unpaired Student's $t$-test (two-tailed), $p = 0.0022$/**H:** $n = 12$ for PE5max, $n = 11$ biologically independent samples for PE5stem. Unpaired Student's $t$-test (two-tailed), $p < 0.0001$/**J:** $n = 3$, unpaired Student's $t$-test (two-tailed), $p = 0.2378$.

Figure S3A: $n = 3$ biologically independent samples, two-way ANOVA; Time x Column Factor $F_{(19,76)} = 1.925$, $p = 0.0239$./**B:** $n = 3$ biologically independent samples, One-way ANOVA, $p = 0.0001$; Dunnett's post-hoc test $p$-values corrected for multiple comparisons: PE2 vs. PE2DD Adjusted $p = 0.0475$, PE2DD vs. PE4 Adjusted $p = 0.0007$.

Figure 4B: $n = 3$ biologically independent samples for every group, One-way ANOVA $p < 0.0001$; Dunnett's post-hoc test $p$-values corrected for multiple comparisons: Mock vs. Cas9 Adjusted $p < 0.0001$, Cas9 vs. PE2 Adjusted $p = 0.0010$, PE2 vs. PE2 + DD Adjusted $p = 0.4796$/**D:** $n = 9$ for Mock and $n = 12$ for every other group, one-way ANOVA $p < 0.0001$; Dunnett's post-hoc test $p$-values corrected for multiple comparisons: Mock vs. siUNG Adjusted $p < 0.0001$, Mock vs. p53DD Adjusted $p = 0.9227$, Mock vs. BE4DI Adjusted $p < 0.0001$./**E:** $n = 9$ biologically independent samples for Mock and $n = 12$ biologically independent samples for siUNG, p53DD and BE4DI, One-way ANOVA $p < 0.0001$; Dunnett's post-hoc test $p$-values corrected for multiple comparisons; Mock vs. siUNG Adjusted $p < 0.0001$, Mock vs. p53DD Adjusted $p = 0.8158$, Mock vs. BE4DI Adjusted $p < 0.0001$/Left panel of G (C to T): $n = 12$ biologically independent samples for AncBE4max and $n = 11$ biologically independent samples for AncBE4stem, unpaired Student's $t$-test (two-tailed), $p = 0.0045$& Right panel of G(C to G): $n = 12$ biologically independent samples for AncBE4max, $n = 11$ biologically independent samples for AncBE4stem, unpaired Student's $t$-test (two-tailed), $p < 0.0001$/**H:** $n = 4$ for PE2 and PE4, unpaired Student's $t$-test (two-tailed), $p = 0.0435$/**K:** $n = 12$ biologically independent samples for both group, unpaired Student t-test (two-tailed), $p < 0.0001$.

Figure S5B $n = 15$ biologically independent samples for Mock, $n = 23$ biologically independent samples for PE and PE2 + DD, one-way ANOVA $p < 0.0001$; Dunnett's post-hoc test $p$-values corrected for multiple comparisons, Mock vs. PE Adjusted $p < 0.0001$, Mock vs. PE2 + DD Adjusted $p = 0.0007$, PE vs. PE2 + DD Adjusted $p = 0.7055$.

## Reporting summary

Further information on research design is available in the Nature Portfolio Reporting Summary linked to this article.

## Data availability

The High-throughput sequencing data generated in this study have been deposited in the NCBI Sequence Read Archive database under accession code BioProject #PRJNA1042664. The RNA-seq data generated in this study have been deposited in the Gene Expression Omnibus under accession code GSE247589. The plasmids encoding AncBE4stem and PE4stem generated in this study have been deposited in the Addgene under accession code no.208766(AncBE4stem) and no. 208768(PE4stem) [https://www.addgene.org]. The analysis pipeline for the analysis of substitution is available at https://github.com/iamleohwang/BE_PE_Sub_analyze. Source data and raw data generated in this study have been deposited in the article repository on Figshare (https://doi.org/10.6084/m9.figshare.23300684) and the Source Data file. Source data are provided in this paper.

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

## Acknowledgements

This research was supported by the National Research Foundation of Korea (NRF) no. RS-2023-00218543 and a grant from the Korean Fund for Regenerative Medicine funded by the Ministry of Science and ICT, and the Ministry of Health and Welfare no. RS-2022- 00070316 and a grant (22202MFDS127) from the Ministry of Food and Drug Safety in 2022-2024.

## Author contributions

H.J.C. conceived the overall study design and led the experiments. J.C.P. and Y.J.K. conducted the experiments and critical discussion of the results. G.H.H., C.Y.K. and S.B. conducted sequencing and data analysis. All authors contributed to the manuscript writing and revising and endorsed the final manuscript.

## Competing interests

The authors declare no competing interests.

## Ethical approval

All authors in this study have met the authorship criteria required by Nature Portfolio journals as their participation was important for the study design and execution.
