## [Peer Review File · Nature Communications]

Reviewers' Comments:

Reviewer #1:

Remarks to the Author:

The paper by Park et al. explores the dual inhibition of p53-mediated cell death and distinct activation of the DNA damage repair system in hPSCs upon DNA damage using CBE or PE. They show that this approach leads to an additive enhancement of editing efficiency. Specifically, the dual inhibition of UNG and p53 through UNG inhibitor (UGI) and p53DD improves CBE outcomes, and the dual inhibition of MMR and p53 using MLH1dn and p53DD further improves PE outcomes. The topic is interesting, and the editing tools developed in this study would benefit the field to improving the editing outcomes in hPSCs. However, the current data presented in the work is far from conclusive, and the inclusion of substantial data sets will be necessary to provide stronger support.

1. The author generated a BE4stem system by adding one more copy of UGI and a p53DD to the BE4max system. The original CBE already contains 2 copies of UGI for UNG inhibition. Why adding a third copy of UGI at the 3' of p53DD? If additional copies of UGI further improve CBE editing outcomes, the author needs to include relevant data.

2. The author showed that inhibition of UNG and p53 improve base editing efficiency meanwhile decrease unwanted edits caused by CBE. For the evaluation of unwanted edits, the author only sequences the target site region. Besides guided off-target events, Cytosine base editors were reported to induce genome and transcriptome-wide cytosine deamination in the edited cells. To evaluate the off-target effects of the inhibition of UNG and p53, comparing genome-wide and transcriptome-wide off-target events by BE4DI and BE4stem is necessary to reach a conclusion that BE4DI and BE4stem increase the edited product purity.

3. In order to demonstrate the combined effects of the MMR and p53 inhibitors, the author needs to include control groups with single inhibition of MMR and p53.

4. How does the combinations of the two inhibitors affect off-target activities by prime editing?

5. Figure 2 showed prime editing improvement in on-target efficiency by using PE4/5stem to conduct GTA and CTT insertions in the genome. Prime editing was reported to create SNPs, insertions, and deletions. How's the PE4/5stem's editing effect for substitutions and deletions? Data for evaluating substitutions and deletions outcomes need to be included to have a full view of the prime editing effect.

6. More endogenous loci for prime editing outcomes will be highly recommended. The current pegRNA information table does not match the figure. Please double check.

7. The author showed that in the TP53 mutated cells, the improvement in CBE editing outcomes is not affected by p53DD but still functions through UGI. Similarly, the author need to include data showing whether MLH1dn is still applicable in this type of cells with improved Prime editing efficiency and product purity.

8. Besides TP53 mutate cells, what is the potential of using BE4stem and PE4/5stem in MMR-deficient cells?

9. The author uses hESCs and hiPSCs as the cell model. However, I didn't see the information of the iPS cell lines utilized. The authors should also provide the basic characterizations of the BE4stem and PE4/5stem edited hPSCs. One suggestion is to incorporate basic G-banding karyotyping to provide an overview of any chromosome abnormalities that may arise in the edited hPSCs.

Reviewer #2:

Remarks to the Author:

The manuscript "Enhancing Precise Genome Editing in Human Pluripotent Stem Cells through Dual

Inhibition of DNA Damage Response and Repair Pathways” by Ju-Chan Park and colleagues reports the finding that dual inhibition of p53-dependent DNA damage sensing and downstream repair pathways additively enhance the efficiencies of base-editing and prime-editing in human pluripotent stem cells (hPSCs). The authors use overexpression of dominant negative p53 (p53DD) together with either UGI expression to inhibit base-excision repair in the case of base-editing, or expression of dominant negative MLH1 (dnMLH1). The data confirms what I believe most people would expect, i.e. an additive effect of these two strategies.

General remarks.

Apart from being somewhat repetitive, the manuscript is well written, and the presentation is clear. The created plasmids could be of use, and should be deposited on Addgene. Overall however, the work lacks both originality and substance, and the presentation of the existing data is insufficiently detailed.

Major points.

1. Lack of originality. All strategies to improve BE and PE described here are known, published, and the relevant papers are (commendably!) cited by the authors. Merely combining known strategies is not original enough for a high impact journal, especially not if the combined effect is additive rather than synergistic. The lack of any new biological insight or of a significant technical advance becomes obvious from the conclusion, page 13: “In conclusion, we have developed a streamlined one-vector system for simultaneous inhibition of the DNA damage response following CBE or PE, leading to improved editing outcomes in hPSCs”.

2. The second major issue is lack of substance: The entire paper could be summarized in a single figure with two panels: A) Dual inhibition of p53 and BER additively improves base-editing in p53WT, but not p53 mutant cells. B) dual inhibition of p53 and MMR additively improves prime editing in p53WT cells. The authors seem to try to compensate for the lack of substance by providing explanatory schemes of the biological processes studied. While this is nicely done and a helpful addition for readers less acquainted with the field, these schematics should go into a supplementary figure. The description of these schematics are not results and should not be in the results section.

3. The third issue is that the data that are provided are heavily processed and collapsed into normalized bar graphs, predominantly showing only the desired editing outcomes. This is insufficient. Absolute editing efficiencies need to be reported in the main Figures, and frequencies of all common other editing outcomes need to be shown and discussed.

4. What are the differences in cell viability using the different constructs? Is the improved fractional editing outcome due to increased editing or due to some other mechanism of cell enrichment? A control nucleofection to estimate percentage transfected should be included.

5. Data availability: The raw data (FASTQ files) must be uploaded to a short read archive, including sufficiently detailed metadata to allow independent re-analysis.

6. The plasmids BEStem and PEStem created here should be deposited at Addgene, sizes should be indicated

Specific points.

1. Figure 1B and C: Why does UNG inhibition reduce C-G conversion and indel formation? What is the proposed mechanism for this? This is a potentially interesting finding that could be explored further.

2. BE4stem has an additional UGI that is released, whereas the originally present UGIs are fused to Cas9. Why was an additional UGI included, instead of an UGN shRNA expressing cassette? This worked well in Figure 1C, and would be a more innovative (and potentially more effective!) approach.

3. Figure 2FG: The legend states that mutations and targets are indicated, but this does not seem to be the case.

4. Addgene plasmids should be cited in the recommended way stating the name of the creator, for example CP1012-CBE_{max} was a gift from David Liu (Addgene plasmid # 119801).
5. Reporting Document: The statement says one of the lines used were mycoplasma tested, but not what the outcome was!

Reviewer #3:

Remarks to the Author:

Reviewer #4:

Remarks to the Author:

Precise genome editing plays a crucial role in clinical applications. However, the editing efficiency of genome editing tools in human pluripotent stem cells (hPSCs) remains unsatisfactory. In the current study conducted by Park et al., they improved the editing efficiency of BE4 and PE4 in hPSCs by employing a dual inhibition approach targeting both p53 and base excision repair or mismatch repair pathways. Furthermore, the author utilized their approach and developed the single vector system BE4stem and PE4/5stem, which might be a potential toolbox in therapeutic gene editing application scenarios.

While the topic of this study is interesting, it is important to note that numerous studies have reported that higher editing efficiency often leads to more undesirable outcomes, such as off-target effects, chromosomal translocations, large deletions, viral DNA or plasmid integrations, and other potential genetic alterations. Regrettably, these aspects were not addressed in the current manuscript, leaving them as open questions for the readers. Consequently, the conclusion regarding precise editing cannot be fully supported based on the present results.

Furthermore, it is worth noting that previous studies have already reported on the concept of transient inhibition of p53 enhancing the efficiencies of CBE/PE in human pluripotent stem cells (hPSCs), as well as the improvement of editing efficiency in CBE or PE through the inhibition of key proteins in the base excision repair (BER) pathway, such as UNG, or the mismatch repair (MMR) pathway, such as MLH1. Consequently, the novelty of this research could be considered moderate, as these concepts have been previously published.

The following concerns are listed below:

Major:

1. Precise genome editing consists of two components: desired and undesired editing products. In this study, the authors focused solely on the detection of desired outcomes, disregarding the evaluation of undesired products. It is imperative for the authors to employ alternative methods to investigate the potential off-target activity, translocations, large deletions, and plasmid integrations associated with the dual inhibition method. Additionally, it should be noted that the inactivation of p53, while enhancing editing efficiency, may suppress apoptosis, allowing cells carrying undesired editing outcomes to persist. Consequently, this raises safety concerns regarding the use of this method. Moreover, the authors need to compare the survival rate after the introduction of CBE with/without the p53DD.
2. In line 12 on page 10, the authors state that "TP53 dominant negative mutations are relatively common in hPSCs," which has led to confusion regarding the rationale for including TP53 dominant negative mutations during genome editing in hPSCs. Alternatively, the authors could reorganize the manuscript as follows: by introducing p53 mutation (p53DD), the editing efficiency is improved, and subsequently, the authors utilized p53DD along with the inhibition of repair pathways to enhance precise genome editing.

Minor:

1. Line 20 on page 3, "HDR" should be removed.
2. Line 23 on page 3, the full name of UNG should be provided.
3. Line 13 on page 8, "(a)" should be from Fig. 1B. The author should clarify the citation.

4. Captions for the supplementary figures are missing.
5. The raw data should also be showed in Fig. S1D.
6. Line 13 on page 9, "(a)" should be from Fig. 2A.
7. Please show the p-value in Fig. 2D.
8. Statistics analysis should be performed for Fig. S1A-D.
9. Figures S3B, S3C, and S3D are not cited throughout the article.
10. In lines 192-194, the authors stated that: "As predicted, PE4stem significantly improved the PE outcomes from a single base pair to large indel (Fig. 2F and S2A) or PE5stem (Fig. 2G and S2B)." However, no detailed information about the "single base pair outcome" and the "large indel" were listed in Fig. S2 or text. The authors should fix it.
11. In lines 201-202, the authors stated that: "The effect of TP53 mutations in PE or CBE was examined in multiple TP53 mutant hPSCs (Fig. S3A)." However, the authors only examined CBE but not PE. The authors should also test the performance of PEstem in TP53 mutant hPSCs.

Reviewer #5:

None

Reviewer #1 (Remarks to the Author):

The paper by Park et al. explores the dual inhibition of p53-mediated cell death and distinct activation of the DNA damage repair system in hPSCs upon DNA damage using CBE or PE. They show that this approach leads to an additive enhancement of editing efficiency. Specifically, the dual inhibition of UNG and p53 through UNG inhibitor (UGI) and p53DD improves CBE outcomes, and the dual inhibition of MMR and p53 using MLH1dn and p53DD further improves PE outcomes. **The topic is interesting, and the editing tools developed in this study would benefit the field to improving the editing outcomes in hPSCs.** However, the current data presented in the work is far from conclusive, and the inclusion of substantial data sets will be necessary to provide stronger support.

1. The author generated a BE4stem system by adding one **more copy of UGI** and a p53DD to the BE4max system. The original CBE already contains 2 copies of UGI for UNG inhibition. Why adding a third copy of UGI at the 3' of p53DD? If additional copies of UGI further improve CBE editing outcomes, the author needs to include relevant data.

- **We appreciate reviewer's comment. Since after the development of cytosine base editors (CBE), CBE has been continuously modified for improvement. In particular, UNG, serving as a key negative determinant for CBE editing outcome as depicted in Figure 1A, has been targeted with addition¹ or ectopic expression² of UGI.**
- **As previously described³, average expression of UNG in hESCs (25 lines) was notably high compared to multiple cancer cell lines (708 lines) and normal cells (15 lines) (Addendum Fig. 1). This would explain why CBE (but not ABE) editing outcomes in hESCs are lower than those in cancer cell lines as highlighted in our previous work³.**
- **Thus, the editing outcome of CBE (i.e., C to T and C to G) is affected by UGI expression even in BE4 with two copies of UGI in hESCs [Figure. S5G and H from the previous study³]. Accordingly, additional knockdown of UNG or additional copy of UGI in hESCs was able to significantly improve the editing outcomes of BE4 (Fig. 1D for C to T and 1E for C to G).**

Rebuttal Figure 1

Rebuttal Figure 1 High UNG expression in hESCs for the editing outcomes of CBE (A) Relative expression level of UNG in 25 hESCs, 708 cancer cell lines and 15 normal cells (dataset from nextbio.com) from Addendum Figure 1 (B) the effect of UGI expression on BE4 editing outcomes in hESCs, Δ UGI: lacking UGI, UGI OE: ectopic expression of UGI from Figure S5G and H of the paper from ³, (C) The effect of UNG knockdown (siUNG) on BE4 editing outcomes (D for C to T, E for C to G) in hESCs from Figure 1D and E

2. The author showed that inhibition of UNG and p53 improve base editing efficiency meanwhile decrease unwanted edits caused by CBE. For the evaluation of unwanted edits, the author only sequences the target site region. Besides guided off-target events, Cytosine base editors were reported to induce genome and transcriptome-wide cytosine deamination in the edited cells. To evaluate the off-target effects of the inhibition of UNG and p53, **comparing genome-wide and transcriptome-wide off-target**

events by BE4DI and BE4stem is necessary to reach a conclusion that BE4DI and BE4stem increase the edited product purity.

- As the reviewer kindly mentioned, transcriptome-wide off target is one of the major side effects of CBE and ABE caused due to the undesirable deamination of RNA bases ^{4, 5}. To examine the transcriptome-wide off-target by dual inhibition of UNG and p53, we determined C to U conversion ratio of conventional CBE (AncBE4max) and BE4stem (AncBE4stem) by RNA-seq analysis as described previously ⁶. The RNA C to U conversion of BE4stem was comparable to that of conventional CBE (Fig. 2G). Similarly, the DNA C to T conversion of BE4stem was found to be comparable with that of traditional CBE (Fig. 2H).

Rebuttal Figure 2

(Figure. 2G and H)

Rebuttal Figure 2 Transcriptome-wide off-target by dual inhibition approach A jitter plot of the efficiency of RNA C to U conversion from RNA sequencing (Figure 2G) and DNA C to T conversion from whole-exom sequencing (Figure 2H).

3. In order to demonstrate the combined effects of the MMR and p53 inhibitors, the author needs to include control groups with single inhibition of MMR and p53.

- As per the reviewer' s comment, the effect of single inhibition on the editing efficiency was tested. Consistently, p53 inhibition (DD) on PE2 significantly improved the efficiency. PE editing efficiency by a single inhibition of MMR was simply achieved by comparison of PE4 [PE2 with expression of dominant negative MLH1 (dnMLH1) ⁷] (Fig. S3A).

Rebuttal Figure 3

(Figure. S3B)

Rebuttal Figure 3 Prime Editing efficiency of single inhibition of p53 or MMR, (Figure. S3B)

4. How does the combinations of the two inhibitors affect off-target activities by prime editing?

- As per the reviewer's comment, we examined the effect of the dual inhibition on off-target activities in prime editing as well as CBE. Editing of off-target (OT1, OT2, OT3 and OT4) on CCR5-3 and (OT1, OT2 and OT3) on HEK2 with CBE was not noticeably altered by dual inhibition (Fig. 2E). Next, off-target editing with PE was examined on the potential off-target sites (OT1, OT2, OT3, OT4 and OT5) of HEK3 1-5 deletion. No evident off-target editing was found in the PE4stem (with dual inhibition approach) as well as PE4max (Fig. 3I). Off-target sites were determined based on guide RNA sequence homology as previously described⁸
- According to the recent study to demonstrate the protective role of p53 on large deletion induced by Cas9⁹, we also additionally examined the large deletion of CBE and PE with dual inhibition. No large deletion was manifested by dual inhibition on CBE and PE (Fig. S2C and S3C).

Rebuttal Figure 4

Rebuttal Figure 4 No noticeable Off-target effect of dual inhibition (A) C to T conversion rate in Four off-target sites (OT1 to OT4) on CCR5-3 (left) and three off-target sites (OT1-OT3) on HEK2 (right) (B) Five off-target sites (OT1 to OT5) on HEK3 (C) Deletion more than 100 base pair long in RNF2 target treated with AncBE4max and

AncBE4stem (Fig. 2F) and PE4max and PE4stem (Fig.3J) (D) Integrated Genomics Viewer (IGV) depth image of RNF2 site exposed to AncBE4max or AncBE4stem (up, Fig. S2C) and PE4max or PE4stem (down, Fig. S3C).

5. Figure 2 showed prime editing improvement in on-target efficiency by using PE4/5stem to conduct GTA and CTT insertions in the genome. Prime editing was reported to create SNPs, insertions, and deletions. How's the PE4/5stem's editing effect for substitutions and deletions? Data for evaluating substitutions and deletions outcomes need to be included to have a full view of the prime editing effect.

- As per the reviewer's comment, additional data of the base substitution (T to A in HEK3), insertion (GTA insertion in RNF2 and AttB sequence insertion in ACTB) and deletion (1-5 and 1-15 deletion in HEK3) were included in Figure 3F. As consistently, dual inhibition approach improved the editing efficiency of PE. In particular, insertion of AttB sequence (38 base pair long) and 15 base deletion, of which prime editing was not affected by suppressing MMR according to our previous report ¹⁰, was improved by p53DD in PE5stem.

Rebuttal Figure 5

(Figure. 3F)

Rebuttal Figure 5 Improvement of editing outcomes of PE with dual inhibition approach, base substitution, deletion in HEK3, GTA insertion in RNF2 and AttB sequence insertion in ACTB (Fig. 3F)

6. More endogenous loci for prime editing outcomes will be highly recommended. The current pegRNA information table does not match the figure. Please double check.

- We appreciate the reviewer's careful comment. As per the reviewer's recommendation, the efficiency of PE4/5stem was additionally tested in HEK3, RNF2 and ACTB (Fig. 3F). The table for pegRNA in the supplementary material was updated accordingly.

7. The author showed that in the TP53 mutated cells, the improvement in CBE editing outcomes is not affected by p53DD but still functions through UGI. Similarly, the author need to include data showing whether MLH1dn is still applicable in this type of cells with improved Prime editing efficiency and product purity.

- As per the reviewer' comment, the effect of dnMLH1 on p53 mutant iPSCs (i.e., SES8) was tested with comparison of editing efficiency of PE4 (with dnMLH1) with those of

PE2. Overall, PE4 showed higher editing efficiency of insertion on RNF2 and HEK3 than PE2 (Fig. 4H).

Rebuttal Figure 6

(Figure. 4H)

Rebuttal Figure 6 The effect of dnMLH1 of PE in p53 mutant iPSCs (Figure 4H)

8. Besides TP53 mutate cells, what is the potential of using BE4stem and PE4/5stem in MMR-deficient cells?

- As per the reviewer's comment, editing outcomes of dual inhibition of CBE and PE were closely determined in p53 mutants. As predicted, the improvement of the editing outcomes of CBE (Fig. 4C, D and E) and PE (Fig. 4I and J) by p53 inhibition in p53 mutant iPSCs was not evident.

Rebuttal Figure 7

A

D

E

(Figure 4C, D and E)

B

J

(Figure 4I and J)

Rebuttal Figure 7 The editing outcome of dual inhibition approach of CBE and PE in TP53 mutant iPSCs (A) C to T conversion rate (left) and C to G conversion rate upon dual inhibition in TP53 mutant iPSCs (Fig. 4C, D and E) (B) The editing efficiency of PE with dual approaches (Fig. 4I and J)

- The improved editing outcomes with p53DD resulted from the transient acquisition of resistance to cell death upon DNA damage (induced by CBE or PE) (Figs. 1B and C for CBE and Figs. 3B and C for PE). Thus, the limited effect of p53DD on editing outcomes in p53 mutant iPSCs was observed in Figure 4. We have observed that hESCs with loss of MSH2 (M2KO hESCs) unexpectedly showed the clear survival trait as similar as that of p53 mutant iPSCs. This unexpected phenomenon was consistent with the recent independent study ¹¹ and iPSCs with mutant MLH1 (Personal communication to Dr. Kim Jung-Hyun in Korea National Stem Cell Bank). Thus, we carefully assumed that dual approach with PE4stem would be less effective in the MMR deficient hESCs as loss of MMR eventually would lead to survival benefit to abrogate the effect of transient p53 inhibition.

9. The author uses hESCs and hiPSCs as the cell model. However, I didn't see the information of the iPS cell lines utilized. The authors should also provide the basic characterizations of the BE4stem and PE4/5stem edited hPSCs. One suggestion is to incorporate basic G-banding karyotyping to provide an overview of any chromosome abnormalities that may arise in the edited hPSCs.

- Two iPSC lines, SES8 and BJ-iPSC were fully characterized in the papers cited in the manuscript ^{12, 13}. In particularly, normal G-banding karyotyping of SES8 was already shown previously ¹². As per the reviewer's suggestion, normal karyotyping of BJ-iPSC after PE was shown in Figure S4B.

Rebuttal Figure 8

A

(Figure. S4A)

Rebuttal Figure 8 Normal G-banding Karyotyping of BJ-iPSCs (Figure S4A)

Reviewer #2 (Remarks to the Author):

The manuscript “Enhancing Precise Genome Editing in Human Pluripotent Stem Cells through Dual Inhibition of DNA Damage Response and Repair Pathways” by Ju-Chan Park and colleagues reports the finding that dual inhibition of p53-dependent DNA damage sensing and downstream repair pathways additively enhance the efficiencies of base-editing and prime-editing in human pluripotent stem cells (hPSCs). The authors use overexpression of dominant negative p53 (p53DD) together with either UGI expression to inhibit base-excision repair in the case of base-editing, or expression of dominant negative MLH1 (dnMLH1). The data confirms what I believe most people would expect, i.e. an additive effect of these two strategies.

General remarks.

Apart from being somewhat repetitive, the manuscript is well written, and the presentation is clear. The created plasmids could be of use, and should be deposited on Addgene. Overall, however, the work lacks both originality and substance, and the presentation of the existing data is insufficiently detailed.

- We value the reviewer's overall feedback on the manuscript. In accordance with the reviewer's suggestions, we have deposited the plasmids on Addgene, contributing to the wider scientific community. Furthermore, we acknowledge and take seriously the reviewer's concerns about the manuscript, particularly regarding the perceived insufficiency of results. We are confident that the inclusion of supplementary results in the revised manuscript will significantly strengthen and reinforce our conclusions.
- Considering the unique characteristics of hPSCs, dual inhibition of DNA damage response and DNA damage repair system would be logically valid to achieve additive effect. Unlike cancer cell lines, where the novel gene editing tools are initially tested and examined, normal cells that should be a genuine target of gene editing for therapeutic strategy, react to DNA damage in cell type specific manner. For example, hematopoietic stem cells (HSCs), on which the first ever FDA-approved gene editing therapy is being expected (i.e., exa-cell) are highly susceptible to cell death upon DNA damage due to p53 activation¹⁴ and are less efficient of DNA damage repair¹⁵, which result in the accumulation of DNA damage during aging¹⁶. Considering the key roles of mismatch repair in HSCs in blood-cancer risk¹⁷, the gene editing strategy needs to be selected according to the distinctive cellular characteristics on DNA damage response and DNA damage repair. Thus, we would like to highlight that the understanding the cellular characteristics of target cells is important for safe and effective gene editing in the discussion as follow

“Consequently, the inhibition of p53 emerges as the most effective approach for PE and CBE in cell types highly susceptible to p53-mediated apoptosis following DNA damage. In particular, HSCs, the exclusive cell source for autologous ex vivo stem cell therapy and the first anticipated FDA-approved gene editing therapy product (i.e., exa-cell), exhibit heightened vulnerability to cell death following DNA damage, primarily due to p53 activation^{18, 19} and compromised DNA damage repair²⁰, which leads to the accrual of DNA damage over time²¹. These distinct characteristics of HSCs may underlie the unexpected genotoxic outcomes of CBE and PE in HSCs²². In this context, the sole inhibition of p53 using p53DD represents the optimal strategy for therapeutic gene editing of HSCs in conditions such as sickle cell anemia

²³ or β -thalassemia ²⁴, both of which are subjects of ongoing clinical trials (NCT05456880). The encouraging outcomes in gene corrected HSCs with base editing, have extended the applicability to other cell types such as hepatic progenitors ²⁵, keratinocytes ²⁶ and so on. Therefore, as similar as the case of hPSCs and HSCs, a comprehensive understanding of the DNA damage response and repair capacity of the target cell types needs to be considered before implementing BE or PE.” (page 15, line 8-22)

Major points.

1. Lack of originality. All strategies to improve BE and PE described here are known, published, and the relevant papers are (commendably!) cited by the authors. Merely combining known strategies is not original enough for a high impact journal, especially not if the combined effect is additive rather than synergistic. The lack of any new biological insight or of a significant technical advance becomes obvious from the conclusion, page 13: “In conclusion, we have developed a streamlined one-vector system for simultaneous inhibition of the DNA damage response following CBE or PE, leading to improved editing outcomes in hPSCs”.

- We appreciate the reviewer’s comment. As pointed out by the reviewer, the components to develop the dual inhibition approach have been acknowledged in the literatures. It is important to note that this study originated from two independent studies from our group. The first demonstrated that elevated activity of base excision repair (BER) resulting from high UNG expression negatively influence the editing outcomes of CBE ³. The second highlighted that impact of mismatch repair (MMR) determine the editing outcomes of PE ¹⁰. While we were preparing this manuscript, we found one paper published in Nature Communications to demonstrate the enhanced editing outcomes of BE and PE by p53 inhibition in hPSCs ²⁷, Considering the role of p53 on active cell death in hPSCs ²⁸ and subsequent negative determinant of Cas9 editing efficiency in hPSCs ²⁹, it was obvious to achieve the improvement of editing outcomes by transient inhibition of p53 in hPSCs which was already examined ²⁸. Likewise, while UGI’s role in enhancing CBE efficiency was previously reported, ectopic expression and linkage of UGI with CBE were reported afterward in multiple papers ^{1, 2}. These examples underscore that the innovation in genome editing technology arises not only from the groundbreaking discovery but also from the innovation based on the existing discoveries.
- Additionally, we demonstrated that TP53 mutant hPSCs, which recurrently occur during in vitro culture ³⁰ was enriched by PE (Fig. S5A and Fig. 4K). As the prime editing is aimed to correct the pathogenic mutations of iPSCs from the patients for the future ex vivo stem cell therapy, the possible enrichment of TP53 mutants by PE would be a serious concern. Thus, we further examined whether the transient p53 inhibition with p53DD would interfere in enrichment of p53 mutant iPSCs as normal iPSCs would achieve the transient resistance to DNA damage by PE (Addendum Fig. 3). Unfortunately, transient inhibition of p53 by p53DD with PE2 failed to achieve a significant reduction of enrichment of p53 mutant iPSCs (Fig. S5B). This attempt was

described in the main text and the requirement of p53 sequencing in surviving colony after gene editing, was highlighted in the discussion as follow.

- “Subsequently, the enrichment of p53 mutant hPSCs was assessed following the introduction of PE2 in the co-culture with normal and p53 mutant lines (Fig. S5A). Consistent with the previously report³¹, the surviving clones after PE2 revealed a significant enrichment of p53 mutant hPSCs (Fig. 4K). We next hypothesized that providing a transient survival benefit to normal hPSCs with p53DD would enable them to compete to p53 mutants in the presence of DNA damage induced by PE2, eventually inhibiting the enrichment of p53 mutant hPSCs during gene editing process. Unfortunately, despite multiple attempts the desired effect of p53DD in inhibiting the enrichment of p53 mutant hPSCs after PE2 mediated prime editing was not achieved (Fig. S5B).” (page 13, line 7-15)

Rebuttal Figure 9

Rebuttal Figure 9 The effect of p53DD on the enrichment of p53 mutant iPSCs after PE (A-B) Graphical summary of DNA damage mediated enrichment of hPSCs harboring mutations in *TP53* gene(A), and desired effect of p53DD in the enrichment of *TP53* mutations(B). Normal cells are colored in green, hPSCs with p53 mutation are colored in red. (C) *TP53* gene mutation R175H ratio analyzed by EGFP positive populations by flow cytometry after gene editing.

2. The second major issue is lack of substance: The entire paper could be summarized in a single figure with two panels: A) Dual inhibition of p53 and BER additively improves base-editing in p53WT, but not p53 mutant cells. B) dual inhibition of p53 and MMR additively improves prime editing in p53WT cells. The authors seem to try to compensate for the lack of substance by providing explanatory schemes of the biological processes studied. While this is nicely done and a helpful addition for readers less acquainted with the field, these schematics should go into a supplementary figure. The description of these schematics are not results and should not be in the results section.

- As per the reviewer’s comment, the most of schematics were moved to the supplementary figure. The key conceptual illustrations were shown in the main

figures (Fig. 1A, 2A, 3A and 3E) to help the reader understand the rationale of dual inhibition approach.

- In order to address total three reviewer's comment during revision, additional results were produced to support our findings. These new results added in the revised manuscript would be able to solidify our findings. I feel certain that the additional results would compensate the lack of substance that the reviewer pointed out.

3. The third issue is that the data that are provided are heavily processed and collapsed into normalized bar graphs, predominantly showing only the desired editing outcomes. This is insufficient. Absolute editing efficiencies need to be reported in the main Figures, and frequencies of all common other editing outcomes need to be shown and discussed.

- As per the reviewer's recommendation, we presented the absolute editing efficiencies in the Figure 1D and E, S1E and F, 2B and C, S2A and B, 3D and F, 4C, 4F, 4I and J, S3A and S4D-J

Rebuttal Figure 10

Rebuttal Figure 10 Absolute data of BE4DI (A), BE4stem (B), PE with dual inhibition (C) in WT, PE2 and PE4 (D), BE4DI (E) and BE4stem (F) PE with dual inhibition (G) in p53 mutant iPSCs

4. What are the differences in cell viability using the different constructs? Is the improved fractional editing outcome due to increased editing or due to some other mechanism of cell enrichment? A control nucleofection to estimate percentage transfected should be included.

- As both CBE and PE cause DNA damage to lead to p53 activation, inducing p53 dependent cell death in hESCs, transient p53 inhibition with p53DD noticeably rescued cell death after introduction of CBE (Fig. 1B and C) or PE (Fig. 3C and S3B).
- As described in the method section in detail (page 5-6, line 20-5), electroporation of CBE or PE into hESCs after single cell dissociation, is a standardized protocol³². Unlike the most of cancer cell lines, hESCs undergo massive cell death by just single cell dissociation through ROCK activation so that Y-27632, a ROCK inhibitor should be used for this process to block cell dissociation induced cell death³³. Thus, lack of Y-27632 caused massive cell death even after electroporation of Mock vector (Fig. S1B). As dissociation induced cell death in hESCs occurs in p53 independent manner³⁴, p53DD failed to protect the cell death regardless of Y27632 treatment (Fig. S1C). Consequently, EGFP transfection through electroporation in hESCs exhibited comparable EGFP+ cells (Fig. S1D). These results clearly demonstrated that the positive effect of p53DD on editing outcomes came from the transient inhibition of p53 dependent cell death upon DNA damage in hESCs, but not just transfection process.

Rebuttal Figure 11

Rebuttal Figure 11 The effect of p53DD in cell death. Live cell imaging and SYTOX staining after

CBE and PE **(A)** or after mock transfection with or without Y27632 **(B and C)**. **(D)** Transient expression of EGFP was analyzed by flow cytometry after 16hrs of transfection with control vector (pcDNA) or p53DD vector.

5. Data availability: The raw data (FASTQ files) must be uploaded to a short read archive, including sufficiently detailed metadata to allow independent re-analysis.

- As per the reviewer's comment, we uploaded the raw data to short read archive.
- "High-throughput sequencing data have been deposited in the NCBI Sequence Read Archive database (<https://www.ncbi.nlm.nih.gov/sra>) under accession number GSE248191 (SuperSeries). [token for reviewers : upezmkkefpcpput].

6. The plasmids BEStem and PEStem created here should be deposited at Addgene, sizes should be indicated

- As per the reviewer's comment, we deposited BE4stem and PE4stem vectors to Addgene [Addgene ID: 208766 (pCMV-AncBE4stem) and 208768 (pCMV-PE4stem), Deposit ID: 83258, status: hold for publication].

Specific points.

1. Figure 1B and C: Why does UNG inhibition reduce C-G conversion and indel formation? What is the proposed mechanism for this? This is a potentially interesting finding that could be explored further.

- Cytosine deaminase in CBE deaminates cytosine into uracil forming G:U mismatch. UNG recognizes and excise uracil to cause apurinic/apyrimidinic (AP) site. Due to high UNG expression in hPSCs, various byproducts of CBE (e.g., C to G, C to A or indel) are produced by the formation of the AP site. Thus, inhibition of UNG by UGI or siUNG not only enhanced editing efficiency but also minimized the incidence of AP site and consequence of production of CBE byproducts in cancer cell lines ^{1, 2, 35} and hPSCs ³.
- For clarification of this, the schematic image to be useful for understanding of CBE action remained depicted in Figure 1A. To clarify this, we described it as follow *"Meanwhile, uracil in G:U mismatch is recognized and removed by UNG (Fig. 1A-c), forming an apurinic (AP) site and subsequently resulting in G:C base repair (Fig. 1A-e) and production of C to G/A or indel mutations (Fig. 1A-f)."* (page 9, line 14-18)

2. BE4stem has an additional UGI that is released, whereas the originally present UGIs are fused to Cas9. Why was an additional UGI included, instead of an UGN shRNA expressing cassette? This worked well in Figure 1C and would be a more innovative (and potentially more effective!) approach.

We deeply appreciate the reviewer's insightful comment. Because of the importance of increasing the efficiency of CBE, various strategies have been developed to increase CBE efficiency, and UGI is one of the most frequently adopted elements to increase CBE efficiency in plasmid vector systems^{1, 2}. As previously described ³, average expression of UNG in hESCs (25 lines) was notably high compared to multiple

cancer cell lines (708 lines) and normal cells (15 lines) (Addendum Fig. 1). This would explain why CBE (but not ABE) editing outcomes in hESCs are lower than those in cancer cell lines as highlighted in our previous work ³.

- However, in order to establish one vector system, siRNA was rather inadequate to be engrafted into one plasmid vector system. This led us to substitute UGI for siUNG to establish BE4stem.
- As the reviewer's kind suggestion, we had attempted to add shRNA for UNG into the plasmid vector system based on the siUNG sequence employed in the previous study ³ (Addendum Fig. 2A). Unfortunately, unlike our expectations, shUNG was less efficient to knockdown UNG (Addendum Fig. 2B) and failed to improve the editing outcome of CBE (Addendum Fig. 2C).

Rebuttal Figure 12

Rebuttal Figure 12 The additional UNG inhibition in hESCs due to high UNG expression (A) The effect of UGI expression on BE4 editing outcomes in hESCs, Δ UGI: lacking UGI, UGI OE: ectopic expression of UGI from Figure S5G and H of the paper from ³ (B) Sanger sequencing data of shUNG expressing vector (Addendum Fig. 3A), UNG expression level after shUNG vector transfection (Addendum Fig. 3B) and C to T conversion ratio in HEK3 target by CBE transfected with pRG control vector or shUNG vector (Addendum Fig. 3C)

3. Figure 2FG: The legend states that mutations and targets are indicated, but this does not seem to be the case.

- As per the reviewer's comment, the error has been corrected.

4. Addgene plasmids should be cited in the recommended way stating the name of the creator, for example CP1012-CBEmax was a gift from David Liu (Addgene plasmid # 119801).

- As per the reviewer's recommendation, we stated the name of the creator of each plasmid.

5. Reporting Document: The statement says one of the lines used was mycoplasma tested, but not what the outcome was!

- As per the reviewer's comment, we added mycoplasma test results for BJ-iPSCs (Fig. S4C) and iPSCs-SES8 (Fig. S4D). They were all negative to mycoplasma when tested by PCR.

Rebuttal Figure 13

(Figure. S4C)

Rebuttal Figure 13 The mycoplasma test. BJ-iPSCs (left) and SES8 (right)

Reviewer #3 (Remarks to the Author):

Reviewer #4 (Remarks to the Author):

Precise genome editing plays a crucial role in clinical applications. However, the editing efficiency of genome editing tools in human pluripotent stem cells (hPSCs) remains unsatisfactory. In the current study conducted by Park et al., they improved the editing efficiency of BE4 and PE4 in hPSCs by employing a dual inhibition approach targeting both p53 and base excision repair or mismatch repair pathways. Furthermore, the author utilized their approach and developed the single vector system BE4stem and PE4/5stem, which might be a potential toolbox in therapeutic gene editing application scenarios. While the topic of this study is interesting, it is important to note that numerous studies have reported that higher editing efficiency often leads to more undesirable outcomes, such as off-target effects, chromosomal translocations, large deletions, viral DNA or plasmid integrations, and other potential genetic alterations. Regrettably, these aspects were not addressed in the current manuscript, leaving them as open questions for the readers.

- As per the reviewer's comment, we could not exclude the possibility that dual inhibition of DNA damage response (i.e., DNA damage repair and p53 dependent cell death) in hPSCs upon CBE or PE would lead to undesirable outcomes. In order to avoid this possibility, we robustly examined the off-target (Fig. 2E and 3I), large deletion (Fig. 2F, S2C and 3J, S3C) and genome-wide and transcriptome-wide off-target events (Fig. 2G, 2H and Fig. 3K and L). Accordingly, we feel assured that dual inhibition approach for CBE or PE did not cause additional undesirable outcomes.
- Additionally, we demonstrated that TP53 mutant hPSCs, which recurrently occur during in vitro culture³⁰ was enriched by PE (Fig. S5A and Fig. 4K). As the prime editing is aimed to correct the pathogenic mutations of iPSCs from the patients for the future ex vivo stem cell therapy, the possible enrichment of TP53 mutants by PE would be a serious concern. Thus, we further examined whether the transient p53 inhibition with p53DD would interfere in enrichment of p53 mutant iPSCs as normal iPSCs would achieve the transient resistance to DNA damage by PE (Addendum Fig. 3). Unfortunately, transient inhibition of p53 by p53DD with PE2 failed to achieve a significant reduction of enrichment of p53 mutant iPSCs (Fig. S5B). This attempt was described in the main text and the requirement of p53 sequencing in surviving colony after gene editing, was highlighted in the main text. *“Subsequently, the enrichment of p53 mutant hPSCs was assessed following the introduction of PE2 in the co-culture with normal and p53 mutant lines (Fig. S5A). Consistent with the previously report³¹, the surviving clones after PE2 revealed a significant enrichment of p53 mutant hPSCs (Fig. 4K). We next hypothesized that providing a transient survival benefit to normal hPSCs with p53DD would enable them to compete to p53 mutants in the presence of DNA damage induced by PE2, eventually inhibiting the enrichment of p53 mutant hPSCs during gene editing process. Unfortunately, despite multiple attempts the desired effect of p53DD in inhibiting the enrichment of p53 mutant hPSCs after PE2 mediated prime editing was not achieved (Fig. S5B).”* (page 13, line 7-15)
- *“The strategy for conferring the transient survival benefits to normal counterparts using p53DD failed to yield the intended effect (S5B). Given the challenges in inhibiting p53 mutant enrichment, assessing the p53 status of corrected hPSCs prior to further clinical usage for autologous stem cell therapy is advisable.”*(page 14, line 19-23)

Consequently, the conclusion regarding precise editing cannot be fully supported based on the present results. Furthermore, it is worth noting that previous studies have already reported on the concept of transient inhibition of p53 enhancing the efficiencies of CBE/PE in human pluripotent stem cells (hPSCs), as well as the improvement of editing efficiency in CBE or PE through the inhibition of key proteins in the base excision repair (BER) pathway, such as UNG, or the mismatch repair (MMR) pathway, such as MLH1. Consequently, the novelty of this research could be considered moderate, as these concepts have been previously published.

- We appreciate the reviewer's comment. As pointed out by the reviewer, the components to develop the dual inhibition approach have been acknowledged in the literatures. It is important to note that this study originated from two independent studies from our group. The first demonstrated that elevated activity of base excision repair (BER) resulting from high UNG expression negatively influence the editing outcomes of CBE ³. The second highlighted that impact of mismatch repair (MMR) determine the editing outcomes of PE ¹⁰. While we were preparing this manuscript, we found one paper published in Nature Communications to demonstrate the enhanced editing outcomes of BE and PE by p53 inhibition in hPSCs ²⁷. Considering the role of p53 on active cell death in hPSCs ²⁸ and subsequent negative determinant of Cas9 editing efficiency in hPSCs ²⁹, it was obvious to achieve the improvement of editing outcomes by transient inhibition of p53 in hPSCs which was already examined ²⁸.
- Likewise, while UGI's role in enhancing CBE efficiency was previously reported, ectopic expression and linkage of UGI with CBE were reported afterward in multiple papers ^{1, 2}. These examples underscore that the innovation in genome editing technology arises not only from the groundbreaking discovery but also from the innovation based on the existing discoveries.

The following concerns are listed below:

Major:

1. Precise genome editing consists of two components: desired and undesired editing products. In this study, the authors focused solely on the detection of desired outcomes, disregarding the evaluation of undesired products. It is imperative for the authors to employ alternative methods to investigate the potential **off-target activity, translocations, large deletions, and plasmid integrations** associated with the dual inhibition method.

- We agree to the reviewer's comment that examining potential adverse outcomes stands as a pivotal aspect in the evolution of genome editing tools. To test the dual inhibition approach causes any undesirable editing outcomes, the off-target editing, large deletion, genome-wide off-target and transcriptome-wide off-target were determined.
- As shown in Rebuttal Figure 11, dual inhibition approach for CBE and PE gave only marginal effect on off-target editing (Fig. 2E and Fig. 3I) and large deletion (Fig. 2F and S2C for CBE and Fig. 3J and S3C for PE).
- To assess the transcriptome-wide off-target of the dual inhibition approach, we assessed RNA C to U conversion in transcriptomes treated with CBE (AncBE4max) and BEstem (AncBE4stem) and evaluated RNA base substitution in transcriptomes

treated with PE4max and PE4stem. The analysis revealed that both CBE and BEstem exhibited similar levels of RNA C to U editing (Fig. 2G). Likewise, the extent of RNA base substitution was comparable between PE4max and PE4stem (Fig. 3K).

- We also investigated gRNA-independent genome-wide impact of the dual inhibition. DNA C to T conversion ratio was comparable between AncBE4max and AncBE4stem (Fig. 2H). Likewise, the DNA base substitution of PE4stem was found to be comparable with that of traditional PE4max (Fig. 3L).

Rebuttal Figure 14

Rebuttal Figure 14 Effect of the dual inhibition approach on undesirable editing outcomes (A) Off-target editing of AncBE4max and AncBE4stem (Left, Fig. 2E), PE4max and PE4stem (Right, Fig. 3I) (B) Deletion more than 100 base pair long in AncBE4max and AncBE4stem (Fig. 2F), PE4max and PE4stem (Fig.3J) (C) Deletion more than 100 base pair long in RNF2 target treated with AncBE4max and AncBE4stem (Fig. 2F) and PE4max and PE4stem (Fig.3J) (D) Integrated Genomics Viewer (IGV) depth image of RNF2 site exposed to AncBE4max or AncBE4stem (up, Fig. S2C) and PE4max or PE4stem (down, Fig. S3C). (E) A jitter plot of the RNA C to U conversion (Left, Fig. 2G) and RNA base substitution (Right, Fig. 3K) from transcriptome data (F) A jitter plot of the DNA C to T edits (Left, Fig. 2H) and DNA base substitution (Right, Fig. 3L) from whole exom sequencing data

Additionally, it should be noted that the inactivation of p53, while enhancing editing efficiency, may suppress apoptosis, **allowing cells carrying undesired editing outcomes to persist**. Consequently, this raises safety concerns regarding the use of this method.

- We acknowledge the reviewer's valid concern regarding safety, particularly in the context of the recurrent occurrence of p53 dominant negative mutations in hPSCs. The prevalence of p53 mutant hPSCs in culture, driven by survival benefits³⁰, raises concern about the translational application of Cas9 gene editing for future autologous ex vivo stem cell therapy³⁶.
- To address this concern, we explored the hypothesis that transient p53 inhibition with p53DD might impede the enrichment of p53 mutant hPSCs. This was based on the notion that normal hPSCs could transiently resist the cell death upon DNA damage induced by PE2 (Addendum Fig. 3). While we had hoped that the p53DD approach

would minimize the enrichment of p53 mutant hPSCs, our experimental results as depicted in Figure S5B, unfortunately did not demonstrate a significant reduction.

- We documented this attempt in the main text and emphasized the necessity of p53 sequencing in surviving colonies after gene editing. Despite the outcome, we appreciate the importance of exploring strategies to mitigate the accumulation of p53 mutant hPSCs and will continue to consider alternative approaches in this regard.
- *“We next hypothesized that providing a transient survival benefit to normal hPSCs with p53DD would enable them to compete to p53 mutants in the presence of DNA damage induced by PE2, eventually inhibiting the enrichment of p53 mutant hPSCs during gene editing process. Unfortunately, despite multiple attempts the desired effect of p53DD in inhibiting the enrichment of p53 mutant hPSCs after PE2 mediated prime editing was not achieved (Fig. S5B).”* (page13, line 10-15)

Rebuttal Figure 15

Rebuttal Figure 15 The effect of p53DD on the enrichment of p53 mutant iPSCs after PE(A-B)
 Graphical summary of DNA damage mediated enrichment of hPSCs harboring mutations in *TP53* gene(A), and desired effect of p53DD in the enrichment of *TP53* mutations(B). Normal cells are colored in green, hPSCs with p53 mutation are colored in red. (C) *TP53* gene mutation R175H ratio analyzed by EGFP positive populations by flow cytometry after gene editing.

Moreover, the authors need to compare the survival rate after the introduction of CBE with/without the p53DD.

- As per the reviewer’s comment, the effect of transient p53 inhibition with p53DD on suppressing apoptosis was monitored by live imaging of cell death upon introduction of CBE or PE. As predicted, p53DD significantly suppressed cell death, induced by expression of CBE (Fig. 1B and C) or PE (Fig. 3C and S3B).

Rebuttal Figure 16

Rebuttal Figure 16 Time dependent Cell death after gene editing with or without p53DD SYTOX positive population after transfection of BE4 (A) or PE2 (B) with or without p53DD

2. In line 12 on page 10, the authors state that "TP53 dominant negative mutations are relatively common in hPSCs," which has led to confusion regarding the rationale for including TP53 dominant negative mutations during genome editing in hPSCs. Alternatively, the authors could reorganize the manuscript as follows: by introducing p53 mutation (p53DD), the editing efficiency is improved, and subsequently, the authors utilized p53DD along with the inhibition of repair pathways to enhance precise genome editing.

- We appreciate the reviewer's insightful comment. As per the reviewer's recommendation, we modified the sentence to clarify the rationale for test of p53DD's effect in TP53 mutant cell line as below.

"Correction of pathogenic mutations in patient-derived hPSCs using BE or PE transforms edited hPSCs into promising cell source for autologous ex vivo stem cell therapy³⁶. Nevertheless, the recurrent genetic alterations such as p53 LOF mutations, which confer a survival advantage, leads to the dominance of mutant hPSCs during in vitro culture³⁰. Thus, p53 mutant hPSCs, when co-exists with normal hPSCs, are enriched in the surviving clones after Cas9 due to the acquired resistance to cell death³¹.

The investigation into the competition-winning ability of p53 mutant hPSCs in gene correction was conducted with iPSCs (SES8, established by lenti-viral reprogramming¹²) harboring a p53 mutation at R175, which corresponds to the recurrent p53 mutant site in hPSCs³⁰ (Fig. 4A). The impact of p53DD on cell death induced by PE2 was found to be only marginal (Fig. 4B)." (page 12, line 11-21)

Minor:

1. Line 20 on page 3, “HDR” should be removed.
- As per the reviewer’s comment, homology directed repair is removed.
2. Line 23 on page 3, the full name of UNG should be provided.
- As per the reviewer’s comment, the full name of UNG is provided.
3. Line 13 on page 8, “(a)” should be from Fig. 1B. The author should clarify the citation.
- As per the reviewer’s comment, we clarified the citation of the figure.
4. Captions for the supplementary figures are missing.
- As per the reviewer’s comment, captions for the supplementary figures are added.
5. The raw data should also be showed in Fig. S1D.
- As per the reviewer’s comment, raw data of Fig. S1D is presented (Fig. 2B).
6. Line 13 on page 9, “(a)” should be from Fig. 2A.
- As per the reviewer’s comment, the citation of the figure is modified.
7. Please show the p-value in Fig. 2D.
- As per the reviewer’s comment, the error is corrected
8. Statistics analysis should be performed for Fig. S1A-D.
- As per the reviewer’s comment, statistics analysis of was performed in normalized data of Fig. S1A-D (Fig. 1D and E, S1E and F).

Rebuttal Figure 17

D

E

D

E

Rebuttal Figure 17 Effect of the dual inhibition on editing outcome of CBE. C to T (Fig. 1D), C to G (Fig. 1E), C to A (Fig. S1D) and insertion and deletion (Fig. S1E) ratio in designated target by AncBE4max

9. Figures S3B, S3C, and S3D are not cited throughout the article.

- As per the reviewer's comment, the error is corrected

10. In lines 192-194, the authors stated that: "As predicted, PE4stem significantly improved the PE outcomes from a single base pair to large indel (Fig. 2F and S2A) or PE5stem (Fig. 2G and S2B)." However, no detailed information about the "single base pair outcome" and the "large indel" were listed in Fig. S2 or text. The authors should fix it.

- As per the reviewer's comment, the error is corrected as follow. "A single vector system, introducing p53DD on PE4 or PE5 (i.e., PE4stem or PE5stem respectively) for dual inhibition of the DNA damage response with p53DD and PE-specific DNA damage repair (i.e., MMR) with dnMLH1 (Fig. 3E), noticeably improved the PE outcomes from single nucleotide

conversion (T to A), deletion (1-5 del and 1-15 del), three nucleotide insertion (GTA ins) and 38 nucleotide (AttB) insertion (Fig. 3F).” (page 11, line 24- page 12, line 4)

11. In lines 201-202, the authors stated that: “The effect of TP53 mutations in PE or CBE was examined in multiple TP53 mutant hPSCs (Fig. S3A).” However, the authors only examined CBE but not PE. The authors should also test the performance of PEstem in TP53 mutant hPSCs.

- We appreciate the reviewer’s careful comment. As per the reviewer’s comment, we tested the effect of dual inhibition as well as PE with p53DD in TP53 mutant hPSCs. Similar with CBE, co-expression of p53DD showed comparable prime editing efficiency with control group (Fig. 4I). Similarly, PE4stem or PE5stem failed to enhance the editing efficiency compared to PE4max or PE5max in p53 mutant iPSCs (Fig. 4J).

Rebuttal Figure 18

Rebuttal Figure 18. Effect of the dual inhibition on editing outcome of prime editing in TP53 mutant cells. Prime editing efficiency of PE4max, PE4max with p53DD, PE5max and PE5max with p53DD (Left, Fig. 4I) and PE4max, PE4stem, PE5max and PE5stem (Right, Fig. 4J)

References

1. Komor AC, *et al.* Improved base excision repair inhibition and bacteriophage Mu Gam protein yields C:G-to-T:A base editors with higher efficiency and product purity. *Sci Adv* **3**, eaao4774 (2017).
2. Wang L, *et al.* Enhanced base editing by co-expression of free uracil DNA glycosylase inhibitor. *Cell Res* **27**, 1289–1292 (2017).
3. Park JC, *et al.* High expression of uracil DNA glycosylase determines C to T substitution in human pluripotent stem cells. *Mol Ther Nucleic Acids* **27**, 175–183 (2022).
4. Zhou C, *et al.* Off-target RNA mutation induced by DNA base editing and its elimination by mutagenesis. *Nature* **571**, 275–278 (2019).
5. Grunewald J, *et al.* Transcriptome-wide off-target RNA editing induced by CRISPR-guided DNA base editors. *Nature* **569**, 433–437 (2019).
6. Jeong YK, *et al.* Adenine base editor engineering reduces editing of bystander cytosines. *Nat Biotechnol* **39**, 1426–1433 (2021).
7. Chen PJ, *et al.* Enhanced prime editing systems by manipulating cellular determinants of editing outcomes. *Cell* **184**, 5635–5652 e5629 (2021).
8. Kim KT, *et al.* Safe scarless cassette-free selection of genome-edited human pluripotent stem cells using temporary drug resistance. *Biomaterials* **262**, 120295 (2020).
9. Cullot G, *et al.* Cell cycle arrest and p53 prevent ON-target megabase-scale rearrangements induced by CRISPR-Cas9. *Nat Commun* **14**, 4072 (2023).
10. Park J-C, *et al.* MutS α and MutS β as size-dependent cellular determinants for prime editing in human embryonic stem cells. *Molecular Therapy-Nucleic Acids*, (2023).
11. Madden-Hennessey K, Gupta D, Radecki AA, Guild C, Rath A, Heinen CD. Loss of mismatch repair promotes a direct selective advantage in human stem cells. *Stem cell reports* **17**, 2661–2673 (2022).
12. Lee TH, *et al.* Functional recapitulation of smooth muscle cells via induced pluripotent stem cells from human aortic smooth muscle cells. *Circ Res* **106**, 120–128 (2010).
13. Bang JS, *et al.* Optimization of episomal reprogramming for generation of human induced pluripotent stem cells from fibroblasts. *Anim Cells Syst (Seoul)* **22**, 132–139 (2018).

14. Dorset SR, Bak RO. The p53 challenge of hematopoietic stem cell gene editing. *Mol Ther Methods Clin Dev* **30**, 83–89 (2023).
15. Biechonski S, Yassin M, Milyavsky M. DNA–damage response in hematopoietic stem cells: an evolutionary trade–off between blood regeneration and leukemia suppression. *Carcinogenesis* **38**, 367–377 (2017).
16. Beerman I, Seita J, Inlay MA, Weissman IL, Rossi DJ. Quiescent hematopoietic stem cells accumulate DNA damage during aging that is repaired upon entry into cell cycle. *Cell stem cell* **15**, 37–50 (2014).
17. Qing Y, Gerson SL. Mismatch repair deficient hematopoietic stem cells are preleukemic stem cells. *PLoS One* **12**, e0182175 (2017).
18. Komarov PG, *et al.* A chemical inhibitor of p53 that protects mice from the side effects of cancer therapy. *Science* **285**, 1733–1737 (1999).
19. Lotem J, Sachs L. Hematopoietic cells from mice deficient in wild–type p53 are more resistant to induction of apoptosis by some agents. *Blood* **82**, 1092–1096 (1993).
20. Biechonski S, *et al.* Attenuated DNA damage responses and increased apoptosis characterize human hematopoietic stem cells exposed to irradiation. *Sci Rep* **8**, 6071 (2018).
21. Rube CE, *et al.* Accumulation of DNA damage in hematopoietic stem and progenitor cells during human aging. *PLoS One* **6**, e17487 (2011).
22. Fiumara M, *et al.* Genotoxic effects of base and prime editing in human hematopoietic stem cells. *Nat Biotechnol*, (2023).
23. Newby GA, *et al.* Base editing of haematopoietic stem cells rescues sickle cell disease in mice. *Nature* **595**, 295–302 (2021).
24. Badat M, *et al.* Direct correction of haemoglobin E beta–thalassaemia using base editors. *Nat Commun* **14**, 2238 (2023).
25. Kim Y, *et al.* Adenine base editing and prime editing of chemically derived hepatic progenitors rescue genetic liver disease. *Cell stem cell* **28**, 1614–1624 e1615 (2021).
26. Hong SA, *et al.* Therapeutic base editing and prime editing of COL7A1 mutations in recessive dystrophic epidermolysis bullosa. *Mol Ther* **30**, 2664–2679 (2022).
27. Li M, *et al.* Transient inhibition of p53 enhances prime editing and cytosine base–editing

- efficiencies in human pluripotent stem cells. *Nat Commun* **13**, 6354 (2022).
28. Liu JC, *et al.* High mitochondrial priming sensitizes hESCs to DNA-damage-induced apoptosis. *Cell stem cell* **13**, 483–491 (2013).
 29. Ihry RJ, *et al.* p53 inhibits CRISPR–Cas9 engineering in human pluripotent stem cells. *Nat Med* **24**, 939–946 (2018).
 30. Merkle FT, *et al.* Human pluripotent stem cells recurrently acquire and expand dominant negative P53 mutations. *Nature* **545**, 229–233 (2017).
 31. Enache OM, *et al.* Cas9 activates the p53 pathway and selects for p53-inactivating mutations. *Nat Genet* **52**, 662–668 (2020).
 32. Park JC, Kim KT, Jang HK, Cha HJ. Transition Substitution of Desired Bases in Human Pluripotent Stem Cells with Base Editors: A Step-by-Step Guide. *Int J Stem Cells*, (2023).
 33. Watanabe K, *et al.* A ROCK inhibitor permits survival of dissociated human embryonic stem cells. *Nat Biotechnol* **25**, 681–686 (2007).
 34. Ohgushi M, *et al.* Molecular pathway and cell state responsible for dissociation-induced apoptosis in human pluripotent stem cells. *Cell stem cell* **7**, 225–239 (2010).
 35. Komor AC, Kim YB, Packer MS, Zuris JA, Liu DR. Programmable editing of a target base in genomic DNA without double-stranded DNA cleavage. *Nature* **533**, 420–424 (2016).
 36. Park JC, *et al.* Gene editing with 'pencil' rather than 'scissors' in human pluripotent stem cells. *Stem Cell Res Ther* **14**, 164 (2023).

Rebuttal Figure 1

(Addendum Fig. 1)

(Figure S5G and H from Park J.C et al, Mol. Ther. Nucleic. Acids)

C

D (Figure 1D and E)

E

Rebuttal Figure 2

(Figure. 2G and H)

Rebuttal Figure 3

Rebuttal Figure 4

A

B

(Figure. 2E)

(Figure. 3I)

C

(Figure 2F)

(Figure 3J)

D

(Figure. S2C)

C

(Figure. S3C)

Rebuttal Figure 5

Rebuttal Figure 6

H

(Figure. 4H)

Rebuttal Figure 7

A

D

E

(Figure 4C, D and E)

B

J

(Figure 4I and J)

Rebuttal Figure 8

A

(Figure. S4A)

Rebuttal Figure 9

Rebuttal Figure 10

A

D

(Figure 1D and E)

E

F

(Figure S1E and F)

B

B

C

(Figure 2B and C)

A

B

(Figure S2A and B)

Rebuttal Figure 11

A

(Figure 1B and C)

B

C

C

(Figure 3C)

B

(Figure S3B)

B

(Figure S1B and C)

C

(Figure S1D)

D

Rebuttal Figure 12

(Addendum Fig. 1)

(Figure S5G and H from Park J.C et al, Mol. Ther. Nucleic. Acids)

Rebuttal Figure 13

C

(Figure. S4C)

Rebuttal Figure 14

D

(Figure. 2G)

(Figure. 3K)

E

(Figure. 2H)

(Figure. 3L)

Rebuttal Figure 15

Rebuttal Figure 16

A

(Figure 1B and C)

B

C

B

C (Figure 3C)

B (Figure S3B)

Addendum Figure 1

Addendum Figure 2

A

B

C

Addendum Figure 3

Reviewers' Comments:

Reviewer #1:

Remarks to the Author:

In the revised manuscript, the authors provided additional data and the revised manuscript has been significantly improved. However, there are still some points that need to be addressed.

1. There are a few questions about the off-target effect studies the author added in the revision.
 - 1) No method can be found related to whole exome sequencing, and data analysis.
 - 2) The author needs to clarify the method of data generation, including the type of samples used for interpreting the data. Were they obtained from a single editing case or multiple different editing cases? Which editing case(s)?
 - 3) Additionally, it looks like the author used a cell mixture right post-electroporation for these experiments. If the on-target efficiency is not high enough, it becomes challenging to compare the unwanted editing frequency using this cell mixture. This is because wildtype cells will significantly dilute the edited cells. The author can consider using the enriched edited cells or edited clonal lines for study.
 - 4) Only jitter plots were shown but the quantification of the edits is required to claim the conversion of BE4stem was comparable to that of conventional CBE.
2. For the C>T base editing, multiple Cs are within the editing window based on the targeting sequence table. The author needs to quantify each change within the target site accordingly.
3. Nicking sgRNA sequence should also be provided in the table.
4. The author provided G-banding Karyotyping of parental BJ-iPSCs (Figure S4A). However, to address the concern of chromosome abnormalities that may arise in the edited cells, the author should present multiple edited clonal lines generated by PEstem tool.

Reviewer #2:

Remarks to the Author:

The revised manuscript is improved, and the most important concerns of the referees have been, in my opinion, satisfactorily addressed.

My main concern however remains: All strategies to improve BE and PE described here are known and published. While of moderate interest to the community, combining these strategies to observe an additive effect does not offer any novel biological insight. I thus remain unconvinced that the manuscript is of sufficient originality and general interest to warrant publication in Nature Communications.

Reviewer #3:

Remarks to the Author:

Reviewer #4:

Remarks to the Author:

The authors have included an elaborate response to the reviewers' comments and have addressed most of my original concerns.

REVIEWER COMMENTS

Reviewer #1 (Remarks to the Author):

In the revised manuscript, the authors provided additional data and the revised manuscript has been significantly improved. However, there are still some points that need to be addressed.

1. There are a few questions about the off-target effect studies the author added in the revision.

1) No method can be found related to whole exome sequencing, and data analysis.

- According to the previous review's comment, we demonstrated that no clear off-target effect occurred by the dual inhibition approach with WGS data. As per the reviewer's comment, the missing detail was added in the revised version. (From Page 7, line 6 to Page 8, line 7)

2) The author needs to clarify the method of data generation, including the type of samples used for interpreting the data. Were they obtained from a single editing case or multiple different editing cases? Which editing case(s)?

- As per the reviewer's comment, the editing cases for off-target analysis were updated in the figure legends. In detail, we have examined the off-target effect of our dual inhibition approach and presented the results following. The information below was updated in the figure legend, corresponding the figure.

Figure #	Targets	Off-target sequence	Number of samples
Figure 2C	HEK2, HEK4, CCR5-3, HPRT1-E4, HPRT1-E6, RNF2	C to G	N = 3
Figure 2E	4 off-target sites of CCR5-3, 3 off-target sites of HEK2	C to T in off-target sequences	N = 3
Figure 2F	RNF2	Large deletion (Deletion > 100 bp)	N = 3
Figure 2G	Whole transcriptome	C to U	N = 1
Figure 2H	Whole exome	C to T	N = 1
Figure 3I	5 off-target sites of HEK3	Insertion and deletion	N = 3
Figure 3J	RNF2	Large deletion (Deletion > 100 bp)	N = 3
Figure 3K	Whole transcriptome	Base substitution	N = 1
Figure 3L	Whole exome	Base substitution	N = 1
Figure S3H	5 off-target sites of RNF2 and HEK3	Insertion and deletion	N = 1

Rebuttal Figure 1 The sample size information of each figure. The information is added in the corresponding figure legend.

3) Additionally, it looks like the author used a cell mixture right post-electroporation for these experiments. If the on-target efficiency is not high enough, it becomes challenging to compare the unwanted editing frequency using this cell mixture. This is because wildtype cells will significantly dilute the edited cells. The author can consider using the enriched edited cells or edited clonal lines for study.

- We are grateful for the reviewer's perceptive observations on the critical aspect of gRNA-dependent off-target editing, which indeed correlates with on-target editing efficiency as noted

previously ¹. Recognizing the importance of thoroughly investigating potential unintended off-target effects in our dual inhibition approach, we have taken proactive steps to address this concern.

- To rigorously assess the specificity of our editing strategy, we established two independent human embryonic stem cell (hESC) clones using the PE4stem system, specifically targeting RNF2_GTA insertion and HEK3_1-10 deletion. To ensure an accurate comparison, we utilized an unedited single clone as a control (Mock) alongside our edited clones (Edit#1 and Edit#2), as depicted in Figure S3C.
- Our analysis demonstrated nearly 100% editing efficiency in the selected clones post-PE4stem application, as shown in Figure S3D. Most importantly, our comprehensive examination revealed no evidence of gRNA-dependent off-target editing at any of the five potential off-target sites, as illustrated in Figures S3E.
- This thorough investigation underscores our commitment to advancing gene editing technologies that are not only efficient but also precise and safe, minimizing the risk of unintended genomic modifications. We believe these findings significantly bolster the reliability and potential clinical applicability of our editing approach, addressing the critical concerns raised.

Rebuttal Figure 2

(Figure S3F, G and H)

Rebuttal Figure 2 gRNA-dependent off-target from the clonally isolated single clones (Edit#1 and Edit#2) compared to unedited clone (Mock)

4) Only jitter plots were shown but the quantification of the edits is required to claim the conversion of BE4stem was comparable to that of conventional CBE.

- Following the recommendations of the reviewer, we induced the quantitative bar graphs that accompany the jitter plots, specifically comparing the putative off-target of base editors (BE) with BE4stem and prime editors (PE4) with PE4stem in Figures S2F and G (for BE comparisons) and Figures S3H and I (for PE comparisons). The data demonstrate that the dual inhibition approach yields effects that are comparable to, if not surpassing, those achieved by traditional methods.

Rebuttal Figure 3

Rebuttal Figure 3 The quantification graph of Jitter plots (Fig. S2F and G, and Fig. S3H and I).

2. For the C>T base editing, multiple Cs are within the editing window based on the targeting sequence table. The author needs to quantify each change within the target site accordingly.

- We are thankful for the reviewer's astute observation regarding the presence of two cytosines within the editing window of the HEK2 and HEK4 targets, which are highlighted in red in Figure S1E. As highlighted, our initial findings showed that the C to T editing efficiency by BE4 was notably higher at the 6th cytosine (C6) for HEK2 and the 5th cytosine (C5) for HEK4, in comparison to the alternate cytosines at positions C4 for HEK2 and C8 for HEK4, respectively. This differential editing efficiency guided our initial focus on C6 of HEK2 and C5 of HEK4 for examining the effects of our dual inhibition approach, as detailed in Figures 1D, E, and Figures 2B, C of the manuscript.
- Responding to the reviewer's insightful suggestion, we extended our analysis to also evaluate the impact of the dual inhibition strategy on the less efficiently edited cytosines, C4 of HEK2 and C8 of HEK4. This comprehensive examination allowed us to ascertain that the dual inhibition approach indeed enhanced editing outcomes across these sites as well, leading to higher C to T conversion rates and reduced C to G editing incidences. These additional findings have been incorporated into the revised manuscript and are presented in Figures S1F, S1G, S2A, S2B, and S4I.
- These results collectively affirm the efficacy of the dual inhibition strategy in improving base editing outcomes, not only for the primary targets (C6 of HEK2 and C5 of HEK4) but also for the secondary cytosines (C4 of HEK2 and C8 of HEK4).
- These results were included in the revised manuscript and the corresponding description was added in the main text (page 11, line 11-15).

Rebuttal Figure 4

Rebuttal Figure 4 Effect of the dual inhibition on multiple Cs in editing window (A) Sequence information of targets, Bases in the editing window are colored in red and target Cs are bolded (left). The C to T editing efficiency of primary and second sites with BE4 was shown in bar graph. C to T, C to G and C to A substitution ratio of C4 in HEK2 and C8 in HEK4 target (B) C to T, C to G and C to A substitution ratio of C4 in HEK2 and C8 in HEK4 target by AncBE4max or AncBE4stem.

3. Nicking sgRNA sequence should also been provided in the table.

- As per the reviewer's recommendation, the sequence information for nicking sgRNA is provided at 1. Sequence information of the supplementary materials.

4. The author provided G-banding Karyotyping of parental BJ-iPSCs (Figure S4A). However, to address the concern of chromosome abnormalities that may arise in the edited cells, the author should present multiple edited clonal lines generated by PEstem tool.

- In direct response to the reviewer's suggestion, we conducted G-banding karyotyping on edited clones derived from H9 human embryonic stem cells (hESCs with 44+XX) (Fig. S3C). Our analysis confirmed that all clones, including both the Mock (control) and the edited clones (Edit#1 and 2), maintained a normal karyotype of 44+XX (Fig. 3I). This result underscores the genetic stability of the edited clones, an essential aspect of our methodology.

- Furthermore, to comprehensively assess the impact of our editing techniques on the characteristics of the hESCs, we evaluated the pluripotency in the edited clones by measuring the expression levels of key pluripotent marker genes (Fig. 3H), along with assessing alkaline phosphatase activity (Fig. S3F). These analyses collectively demonstrate that our editing approach does not compromise the pluripotency, further evidencing the method's safety and efficacy.

Rebuttal Figure 5

Rebuttal Figure 5 The characterization of edited clones for pluripotency and G-banding karyotype (Fig. 3H and I)

Reviewer #2 (Remarks to the Author):

The revised manuscript is improved, and the most important concerns of the referees have been, in my opinion, satisfactorily addressed.

My main concern however remains: All strategies to improve BE and PE described here are known and published. While of moderate interest to the community, combining these strategies to observe an additive effect does not offer any novel biological insight. I thus remain unconvinced that the manuscript is of sufficient originality and general interest to warrant publication in Nature Communications.

- We sincerely value the reviewer's concerns and thank them for their constructive criticism. As emphasized within our manuscript, the application of Cas9 for knock-in applications, especially in human pluripotent stem cells (hPSCs), often leads to large, unintended deletions. This presents a significant challenge for clinical applications, particularly for ex vivo autologous stem cell therapy targeting genetic diseases. Our approach, utilizing base editors (BEs) and prime editors (PEs) that do not induce double-strand breaks, emerges as a potentially safer and more reliable method for correcting mutations in patient-derived induced pluripotent stem cells (iPSCs). The enhancements we've achieved in BE and PE efficiency, without increasing off-target effects, mark a significant step forward in the gene correction of iPSCs. We are grateful for Reviewer #1's suggestion to rigorously assess the safety of this dual approach, acknowledging its importance in advancing stem cell therapy.
- Furthermore, we agree that ongoing efforts to improve BE and PE editing efficiency in hPSCs are crucial until more innovative strategies are discovered. Our work represents not just a combination of existing technologies but a refined, potentially transformative application with significant implications for gene therapy's future. We believe that our findings contribute valuable insights to the field and merit consideration for publication due to their practical relevance and potential impact.

Reviewer #3 (Remarks to the Author):

Reviewer #4 (Remarks to the Author):

The authors have included an elaborate response to the reviewers' comments and have addressed most of my original concerns.

- We appreciate the positive response from the reviewer.

References

1. Zhang L, He W, Fu R, Wang S, Chen Y, Xu H. Guide-specific loss of efficiency and off-target reduction with Cas9 variants. *Nucleic Acids Res* **51**, 9880-9893 (2023).

Rebuttal Figure 1

Figure #	Targets	Off-target sequence	Number of samples
Figure 2C	HEK2, HEK4, CCR5-3, HPRT1-E4, HPRT1-E6, RNF2	C to G	N = 3
Figure 2E	4 off-target sites of CCR5-3, 3 off-target sites of HEK2	C to T in off-target sequences	N = 3
Figure 2F	RNF2	Large deletion (Deletion > 100 bp)	N = 3
Figure 2G	Whole transcriptome	C to U	N = 1
Figure 2H	Whole exome	C to T	N = 1
Figure 3I	5 off-target sites of HEK3	Insertion and deletion	N = 3
Figure 3J	RNF2	Large deletion (Deletion > 100 bp)	N = 3
Figure 3K	Whole transcriptome	Base substitution	N = 1
Figure 3L	Whole exome	Base substitution	N = 1
Figure S3H	5 off-target sites of RNF2 and HEK3	Insertion and deletion	N = 1

Rebuttal Figure 2

(Figure S3F, G and H)

Rebuttal Figure 3

(Figure S2F and G)

(Figure S3H and I)

Rebuttal Figure 4

A **G**

Target	Spacer (without PAM)	Bystander
CCR5-3	GGC AGCAT AGTGAGCCCAGA	n.s
CCR5-10	GGT GACA AGTGTGATCACTT	n.s
HPRT1-E4	GGG GACATA AAAAGTAATTGG	n.s
sgRNA HPRT1-E6	GTAT AATC CAAAGATGGTCA	n.s
HEK2	GAA C₄AC₆ AAAGCATAGACTGC	C4
HEK4	GGC AC₅TC₈ GGCTGGAGGTGG	C8
RNF2	GTC ATCTT AGTCATTACCTG	n.s

(Figure. S1G, H and I)

B

(Figure. S2C and D)

Rebuttal Figure 5

(Figure S3I)

Reviewers' Comments:

Reviewer #1:

Remarks to the Author:

I appreciate the efforts the author made to address my questions. However, I still have concerns regarding the genome-wide off-target analysis and the information of samples that I hope the authors will address and clarify.

The nucleotide alterations by base editor are distributed throughout the genome and range from less than 1% to more than 80% (e.g. Grünwald et al. 2019, Nature; and in the reference author cited: Jeong YK, et al.2021,Nat Biotechnology). Across the all the jitter plot figures, the authors only provided the data for the nucleotide change up to 30% (Fig. 2G,H, fig 3K,L). The quantification only showed the change ranging from 10 to 30% (supple-fig 2F,G). The author needs to provide the full data in these figures. In addition, I still cannot find the information for what's the samples were used. e.g. what are the samples or targets of AncBE4max and AncBE4 stem treated cells were used in (Fig2G,H, fig 3K,L)?

REVIEWER COMMENTS

Reviewer #1 (Remarks to the Author):

I appreciate the efforts the author made to address my questions. However, I still have concerns regarding the genome-wide off-target analysis and the information of samples that I hope the authors will address and clarify.

The nucleotide alterations by base editor are distributed throughout the genome and range from less than 1% to more than 80% (e.g. Grünewald et al. 2019, Nature; and in the reference author cited: Jeong YK, et al. 2021, Nat Biotechnology). Across the all the jitter plot figures, the authors only provided the data for the nucleotide change up to 30% (Fig. 2G,H, fig 3K,L). The quantification only showed the change ranging from 10 to 30% (supple-fig 2F.G). The author needs to provide the full data in these figures.

- We are grateful for the reviewer's insightful observations. In order to remove background noises in our initial jitter plots from the genome- and transcriptome-wide off-target analysis, we implemented an advanced background filtration technique as outlined in our referenced work ¹. The refinement led to updated jitter plots, which now display a range from 0 to 100%, as depicted in Figures 2G and 3K. Additionally, to enhance our quantification approach, we incorporated the number of nucleotide changes, adhering to the methodology detailed in the same reference ¹. (line 9-13, page 11).
- The methodology section has been updated to include a comprehensive description of the technique employed to generate these enhanced jitter plots, with the appropriate citation provided. This adjustment not only addresses the concerns raised but also significantly improves the clarity and accuracy of our findings (line 19-25, page 7 and line 1-3, page 8).

Rebuttal Figure 1

G

(Figures 2G)

K

(Figures 3K)

Rebuttal Figure 1 The updated Jitter Plots ranging from 0 to 100% (Figure 2G and 3K)

In addition, I still cannot find the information for what's the samples were used. e.g. what are the samples or targets of AncBE4max and AncBE4 stem treated cells were used in (Fig2G,H, fig 3K,L)?

- In accordance with the reviewer's feedback, we have addressed the previously noted omission by incorporating the missing data relevant to Figures 2G and 3K into Figures S2F and S3D, respectively.

Rebuttal Figure 2

F

(Figures S2F)

D

(Figures S3D)

Rebuttal Figure 2 Base editing and prime editing data, correspondent to Figure 2G and 3K

References

1. Gaudelli NM, *et al.* Directed evolution of adenine base editors with increased activity and therapeutic application. *Nat Biotechnol* **38**, 892-900 (2020).

Rebuttal Figure 1

G

	DNA (C to T)		RNA (C to U)	
n	160669	171820	48491	50782

(Figures 2G)

K

	DNA		RNA	
n	540683	489835	139621	180675

(Figures 3K)

Rebuttal Figure 2

(Figures S2F)

(Figures S3D)